

# Technical note: Refining $\delta^{15}N$ isotopic fingerprints of local NOx for accurate source identification of nitrate in PM$_{2.5}$

**Hao Xiao [1,2], Qinkai Li [3], Shiyuan Ding [2], Wenjing Dai [2], Gaoyang Cui [4], Xiaodong Li [2*]**

[1] School of Agriculture and Biology, Shanghai Jiao Tong University, Shanghai, 200240, China;

[2] Institute of Surface-Earth System Science, School of Earth System Science, Tianjin University, Tianjin, 300072, China;

[3] Jiangxi Key Laboratory of Environmental Pollution Control, Jiangxi Academy of Eco-Environmental Sciences and Planning, Nanchang, 330039, China;

[4] The College of Geography and Environmental Science, Henan University, Kaifeng, 475004, China;

\*    Correspondence: Xiaodong Li (xiaodong.li@tju.edu.cn)

● **Abstract**

Stable nitrogen isotopic composition ($\delta^{15}N$) has proven to be a valuable tool for identifying sources of nitrates (NO$_3^-$) in PM$_{2.5}$. However, the absence of a systematic study on the $\delta^{15}N$ values of domestic NOx sources hinders accurate identification of NO$_3^-$ sources in China. Here, we systematically determined and refined $\delta^{15}N$ values for six categories of NOx sources in the local Tianjin area using an active sampling method. Moreover, the $\delta^{15}N$ values of NO$_3^-$ in PM$_{2.5}$ were measured during pre-heating, mid-heating and late-heating periods, which are the most heavily polluted in Tianjin. Results shown that the representative nature and region-specific characteristics of isotopic fingerprints for six categories of NOx sources in Tianjin. The Bayesian isotope mixing (MixSIAR) model demonstrated that coal combustion, biomass burning, and vehicle exhaust collectively contributed more than 60%, dominating the sources of NO$_3^-$ during sampling periods in Tianjin. However, failure to consider the isotopic signatures of local NOx sources could result in an underestimation of the contribution from coal combustion. Additionally, the absence of industrial sources, an uncharacterized source in previous studies, may directly result in the contribution fraction of other sources being overestimated by the model more than 15%. Notably, as the number of sources input to the model increased, the contribution of various NOx sources was becoming more stable, and the inter-influence between various sources significantly reduced. This study demonstrated that the refined isotopic fingerprint in a region-specific context could more effectively distinguish source of NO$_3^-$, thereby providing valuable insights for controlling NO$_3^-$ pollution.

## 1. Introduction

In recent decades, the acceleration of urbanization and modernization in China



has inevitably led to persistent and frequent incidents of atmospheric $PM_{2.5}$ pollution
in urban areas (Zhang et al., 2023;Meng et al., 2024). SNA ($SO_4^{2-}$, $NO_3^-$ and $NH_4^+$)
constitutes one of the most important components of $PM_{2.5}$, and its elevated
concentration will exacerbate the pollution level of $PM_{2.5}$ (Huang et al., 2014). A
series of scientific and effective air pollutant emission control measures has led to a
significant decrease in the concentration of $SO_4^{2-}$ in $PM_{2.5}$ in urban areas of China has
decreased significantly (Wang et al., 2022). However, the concentration and
percentage of $NO_3^-$ in $PM_{2.5}$ have shown a gradual increase (Wang et al., 2022;Zhang
et al., 2020). Previous studies have indicated that $NO_3^-$ has surpassed $SO_4^{2-}$ as the
primary inorganic component of atmospheric $PM_{2.5}$ in the northern Chinese cities,
with mass concentrations accounting for approximately 5% ~ 26% (Zong et al.,
2022b;Zhang et al., 2019;Xie et al., 2019). Consequently, the accurate identification
of the sources of $NO_3^-$ is essential for the development of effective air management
measures, which can effectively control the occurrence of urban haze weather.

The accurate identification of the sources of atmospheric NOx was complicated
by the considerable complexity and diversity of the sources involved, which include,
but are not limited to, coal combustion, vehicle exhaust, biomass burning and soil
emissions (Huang et al., 2017;Duncan et al., 2016). The reliable identification of the
sources of NOx in the atmosphere was achieved using stable nitrogen isotopes
composition ($\delta^{15}N$) (Zong et al., 2017;Song et al., 2021). However, to achieve the
most accurate results, it is essential to accurately identify the $\delta^{15}N$ values of the
atmospheric NOx source (Zhang et al., 2024a;Lin et al., 2021). Although the $\delta^{15}N$
values from some NOx sources have been reported by other studies (Zong et al.,
2020a;Zong et al., 2022a). However, the majority of current research reports on the
$\delta^{15}N$ values of NOx from different sources in the atmosphere originates from foreign
countries, and the collection methods have not been unified (Elliott et al.,
2019;Walters et al., 2015a;Walters et al., 2015b). For instance, Felix and Elliott (2014)
reported the $\delta^{15}N$-NOx from vehicle exhaust as +14.2 ± 1.9‰ based on the active
sampler. However, a relatedly negative value (−11.4 ± 6.9‰) from vehicle exhaust
was reported by the passive sampler (Walters et al., 2018). Similar discrepancies have
also been observed in other NOx sources (Li and Wang, 2008;Elliott et al., 2019).
Furthermore, the production mechanisms of different NOx types can also affect its
$\delta^{15}N$ values, such as fuel type NOx and thermal type NOx (Heaton, 1990). Coal
combustion can result in the release of both fuel type NOx and thermal type NOx, due
to the high combustion temperature and the abundant nitrogen components (Heaton,
1990;Felix et al., 2012). However, biomass burns at low temperatures (250 to
1200 °C), and the process produces mainly fuel NOx, with $\delta^{15}N$ depending on the
relative abundance of nitrogenous organic matter $^{15}N$ in the biomass itself (Zong et al.,
2022a). Therefore, considering the regional differences in the relative abundance of
$^{15}N$ in fuels (Zong et al., 2022a;Shi et al., 2022), these $\delta^{15}N$ values reported abroad for
fuel-based NOx may not be applicable to domestic studies. Furthermore, the previous
studies have not yet provided a comprehensive overview of the $\delta^{15}N$ values of NOx
sources. There is a lack of systematic studies on the $\delta^{15}N$ characteristics of different
NOx sources at both the domestic and international level. For instance, the $\delta^{15}N$





values of NOx from industrial emissions and natural gas combustion in urban areas
have rarely reported. Despite these sources being included in the NOx emission
inventory, the current isotopic fingerprint database lacks a clear definition. In this
context, the use of associated model calculations to quantify the source contribution
of NOx or $NO_3^-$ to the regional atmosphere may introduce a high level of uncertainty
(Zhang et al., 2024a). Therefore, it is of great significance to enhance the existing
$\delta^{15}N$ values of NOx sources.
Tianjin, recognized as one of most heavily polluted cities in China, experiences
strongly influenced from severe haze pollution (Xiao et al., 2022, 2023;Xiao et al.,
2024b). Particularly during heating periods, episodes of haze formation characterized
by abrupt increases in $NO_3^-$ concentrations have been observed (Zou et al., 2018;Feng
et al., 2020). This phenomenon is attributable to the predominant use of coal as the
primary heating fuel in Tianjin, resulting in considerable NOx emissions (Zhao et al.,
2021). Recent research efforts in Tianjin have focused on identifying the sources of
$NO_3^-$ through its dual isotopic compositions (Zhang et al., 2019;Xiao et al., 2023).
Nevertheless, there is a paucity of systematic characterization of the $\delta^{15}N$-NOx source
signatures in Tianjin, and even in China, which contributes to significant uncertainties
regarding NOx source contributions (Zhang et al., 2024a). Consequently, to enhance
understanding of atmospheric $NO_3^-$ sources, it is imperative to refine the $\delta^{15}N$ values
of various local sources in Tianjin.
Here, we established a refined isotopic fingerprint for major NOx sources in
Tianjin area using an active sampling method, which including previously
uncharacterized sources in China such as industrial emissions and natural gas
combustion. To better understand the need for established refined isotopic fingerprint
in a region-specific context for major NOx sources, $PM_{2.5}$ samples were collected in
Tianjin across three distinct periods: pre-heating (29th October 2018 to 25th
November 2018), mid-heating (8th January 2019 to 22nd January 2019), and late-
heating (7th March 2019 to 3rd April 2019). Based on the Bayesian isotope mixing
(MixSIAR) model, this study compares and explains the differences in the source
resolution results due to the $\delta^{15}N$-NOx sources measured locally in Tianjin and those
measured by previous studies. Notably, coal combustion activity was most
pronounced during the mid-heating period compared to other periods. Given the
proposition that coal combustion is a significant source of $NO_3^-$, it is conceivable that
coal combustion would exhibit a notably higher contribution fraction during the mid-
heating period compared to the pre-heating and late-heating periods (Feng et al.,
2020). This provides an opportunity to explore the necessity of establishing local
source $\delta^{15}N$ values for NOx emissions in Tianjin. The study will enhance our
understanding of the source of $NO_3^-$ in $PM_{2.5}$, and further emphasize the necessity for
the establishment of a refined isotopic fingerprint for NOx sources in future studies.





## 2. Materials and Methods

### 2.1 Description of the location and sample collection



PM$_{2.5}$ samples were collected from the rooftop of Tianjin University, located in
the Nankai District, Tianjin (Figure S1). The roof stands approximately 25 m above
ground level, with no discernible sources of pollutants in the vicinity, such as factories
or construction sites. A high-volume air sampler (Tisch, USA) equipped with quartz
fiber filters (Pallflex, 20 × 25 cm) was utilised for PM$_{2.5}$ collection, operating at a
flow rate of 1.05 m$^3$ min$^{-1}$. Prior to sampling, all quartz fiber filters underwent
combustion at 450°C for 6 hours to eliminate potential interference from organic
matter. Samples were collected at two distinct time points: daytime samples between
8:00 and 18:00, and nighttime samples between 18:30 and 7:30 the following morning.
Pollutant concentrations (e.g., PM$_{2.5}$, SO$_2$, NO$_2$, CO) and corresponding
meteorological parameters (e.g., ambient temperature, T; relative humidity, RH; wind
speed, WS) during sampling periods were obtained from nearby monitoring stations
(Xiao et al., 2024b).
To enable comparison, a diaphragm pump (Laoying 3072, Qingdao Laoying
Environmental Science and Technology, China) was uniformly employed to actively
absorb NOx emissions from various sources (Figure S2). Initially, hydrophobic Teflon
membrane (TF-200, Pall, USA) and nylon membrane (BNRG810S, Pall, USA) were
used to exclude interference of particulate NO$_3^-$ and gaseous HNO$_3$ emitted by
sources, respectively. Filtered gases were then passed through an alkaline solution of
0.5 mol L$^{-1}$ NaOH and 0.25 KMnO$_4$, known for its strong oxidizing properties,
oxidizing NOx entering the absorbing solution to form NO$_2^-$ or NO$_3^-$ and store it in
the absorbing vial (Fibiger et al., 2014;Fibiger and Hastings, 2016). The added NaOH
served to react with NOx and enhance the viscosity of the absorbing solution, thereby
improving conversion efficiency (Sada et al., 1977). However, since KMnO$_4$ in the
absorbing solution is in excess, incompletely reacted KMnO$_4$ in the solution must be
further reduced in the laboratory. This study involved 6 emission sources of NOx in
the local Tianjin area, including coal-fired power plants, gas-fired power plants,
biomass burning, vehicle exhaust, iron and steel smelting and soil emission sources
(see Supporting Information Text S1 for detailed description). All collected PM$_{2.5}$
samples and gaseous NOx from various sources were stored at -20 °C in a refrigerator
after sampling. It was worth noting that an additional blank sample was prepared for
each sampling campaign, in parallel with the other samples.

### 2.2 Chemical and isotopic analysis


In the ultraclean room, a proportion of the particulate matter from the each PM$_{2.5}$
sample was cutted and transferred to a 50 ml centrifuge tube. Subsequently, samples
from each filter were extracted using Milli-Q water, which has a demonstrated ionic
strength of 18.2 MΩ cm (Millipore, United States), via ultrasonication and



centrifugation. Ionic chromatography (Dionex Aquion, Thermo Fisher Scientific, Inc.,
Waltham, MA, USA) was employed following established methodologies to
determine the presence of water-soluble major ions (e.g., $SO_4^{2-}$, $NO_3^-$, and $NH_4^+$) in
$PM_{2.5}$ (Xiao et al., 2024b). For the gaseous NOx samples from various sources, an
excess of $H_2O_2$ was added to the absorbent solution until all the incompletely reacted
$KMnO_4$ in the sampled absorbent solution was converted, resulting in the formation of
black $MnO_2$ precipitate (Fibiger and Hastings, 2016). Subsequently, the resulting
suspension was centrifuged at high speed (4000 r min$^{-1}$) for 10 minutes to obtain the
supernatant, which was neutralized using electronic grade hydrochloric acid (HCl) at
a mass concentration of approximately 38% to neutralize under-reacted sodium
hydroxide. Finally, the supernatant was analyzed for $NO_2^-$ and $NO_3^-$ concentrations
on a Skalar San++ continuous flow nutrient salt analyzer. It was noteworthy that the
concentration of $NO_2^-$ in the absorbent solution after treatment with a strong oxidant
(i.e., $H_2O_2$) as described above is typically extremely low, typically below 0.005 μg
mL$^{-1}$.
In this study, we utilized the bacterial denitrification method to determine the
dual isotopic values of $NO_3^-$ ($\delta^{15}N$ and $\delta^{18}O$) in $PM_{2.5}$ and absorbent solution. Further
details can be found in our previous study (Xiao et al., 2023;Li et al., 2021). Briefly,
extracted $NO_3^-$ were quantitatively converted to $N_2O$ through the action of
denitrifying bacteria, namely Pseudomonas aureofaciens, ATCC 13985, which lacked
$N_2O$ reductase (Luo et al., 2020b). Subsequently, the $\delta^{15}N$ and $\delta^{18}O$ values of $N_2O$
were determined using GasBench-II with continuous flow isotope ratio mass
spectrometry (IRMS; Thermo Fisher DELTA V advantage, Thermo Fisher Scientific,
Inc.) for on-line analysis. The study employed three isotopic international standards:
USGS32, USGS34 and IAEA-N3, and the analytical accuracies for both $\delta^{15}N$ and
$\delta^{18}O$ were ± 0.2‰ and ± 0.3‰, respectively. Furthermore, the influence of $NO_2^-$ was
deemed negligible as its concentration fell below 2% of the $NO_3^-$ level in all samples
(Luo et al., 2019).
**2.3 Stable isotopic mixing model**
The Bayesian model enabled the determination of the probability distribution of
the contribution of each source to a mixture (Parnell et al., 2010). Subsequently, a
Bayesian isotope mixing model was implemented in the R software package (Stable
Isotope Analysis in R, SIAR) to estimate potential sources of atmospheric $NO_3^-$ in
$PM_{2.5}$ in this study (Zong et al., 2017). Briefly, the model initiates with establishing a
logistic prior distribution, followed by determining the probability contribution
distribution of each source to the mixture. Further details are available in our previous
study (Xiao et al., 2023;Li et al., 2021). It was noteworthy that an obvious isotopic
fractionation process occurs during the conversion of NOx to $NO_3^-$. Therefore, the
nitrogen isotope fractionation coefficient (εN) resulting from NOx to $NO_3^-$ conversion
should be calculated prior to determining the relative contribution of NOx sources
using MixSIAR model (Text S2). Furthermore, to enhance the reliability of the study
results, the model generated 10,000 potential scenarios for each evaluated potential



source (Song et al., 2019;Fan et al., 2020).

### 3. Results and discussion

#### 3.1 The $\delta^{15}N$ values of major $NO_x$ emission sources

This study sampled three categories of $NO_x$ sources associated with the combustion of fossil fuels, specifically vehicle exhaust, coal combustion, and natural gas combustion. As shown in Figure 1 and Table 1, the NOx concentrations emitted by these sources exceeded atmospheric NOx concentration levels in urban China. The $\delta^{15}N$ values and concentrations of NOx in vehicle exhaust exhibited a range of −18.3 ‰ to 7.9 ‰ and 0.006 to 223.8 ppm, respectively. Furthermore, slight differences were observed between the $\delta^{15}N$ values in vehicle exhaust at different sampling sites (Figure S3), which can be attributed to the varying nitrogen contents of the fuels, the NOx generation pathways and the efficiency of the three-way catalytic devices (Walters et al., 2015a;Zong et al., 2020a;Heaton, 1990). For instance, vehicle exhaust can produce both thermal and fuel NOx. Thermal NOx is generated at high temperatures exceeding 1500°C and is influenced by factors such as the molar concentrations of $O_2$ and $N_2$ and combustion temperature (Walters et al., 2015b). In contrast, fuel NOx is primarily related to the nitrogen content of the fuel (Walters et al., 2015b). Walters et al. (2015a) observed that fuel NOx exhibited more positive $\delta^{15}N$ values than thermal NOx, and catalytic treatment could also yield positive $\delta^{15}N$ values. When compared with previous studies (Figure S4a and Table 1), our results align with domestic reports but significantly differ from foreign studies. This variation can be attributed to significant differences in $\delta^{15}N$ values of oils due to their generation and depositional settings (Williams et al., 1995), indicating local characteristics of $\delta^{15}N$ values of NOx in vehicle exhaust.

The $\delta^{15}N$ value of NOx emissions from the coal-fired power plant (coal combustion) in Tianjin ranged from +9.4 ‰ to +15.4 ‰, with a mean value of +12.3 ± 1.7 ‰ (Figure 1). Our results differ significantly from previous reported $\delta^{15}N$-NOx values from coal combustion (Figure S4b). A previous study found that the $\delta^{15}N$-NOx values from coal combustion were primarily influenced by thermal NOx production related to combustion temperature and NOx reduction technology related to fuel-to-air ratio, rather than by $\delta^{15}N$ values of the coal itself (Felix et al., 2012). However, the coal combustion temperatures of approximately 1300 to 1400 °C observed in power plants in this study were insufficient to produce thermal NOx (Heaton, 1990). Consequently, the $\delta^{15}N$-NOx value from coal combustion in this study may be related to the $^{15}N$ abundance of coal, indicating local characteristics of $\delta^{15}N$ values of NOx from coal combustion. It is noteworthy that a considerable range of values (-31.3 ‰ ~ -17.5 ‰) and a relatively negative mean value (-24.8 ± 5.6 ‰) of $\delta^{15}N$-NOx from natural gas combustion were observed, in comparison to the aforementioned sources (Figure 1). Generally, natural gas has a low nitrogen content, and its combustion primarily produces thermal NOx, with $\delta^{15}N$ values depending on temperature, pressure and oxygen content in the combustion chamber (Heaton, 1990). In this study, the combustion chamber of a gas-fired power plant can have a temperature of more





than 2000 °C, generating NOx mainly through the extended Zeldovich mechanism (Zong et al., 2020a). Consequently, the $\delta^{15}N$ values of NOx from natural gas combustion exhibit a significantly more negative trend than those observed in the two aforementioned sources. However, our results were more negative than those reported by Walters et al. (2015b) for NOx emitted from residential gas furnaces in Indiana, USA ( ranging from -19.7 ‰ to -13.9 ‰ and -16.5 ± 1.7 ‰). This discrepancy can be attributed to the combustion process employed in this study, which necessitates the introduction of sufficient air into the combustion chamber. To achieve this, 'fresh air' was introduced into to the chamber, which may have resulted in a reduction in $^{14}N^{14}N$ loss and the generation of negative $\delta^{15}N$-NOx values in this study compared to those observed in previous studies.

NOx emissions from industrial sources, such as the iron and steel industry, arise from various processes including sintering, pelletizing, and hot blast furnaces (Wang et al., 2019;Zhao et al., 2017). Generally, the $\delta^{15}N$-NOx value from industrial emission sources differs significantly from those of emissions from fossil fuel combustion (Figure 1b), emphasizing the representativeness of the isotopic fingerprint in industrial emission sources. The $\delta^{15}N$ values of NOx emitted from the hot blast furnace were $-43.1 \pm 12.3$ ‰, in contrast to the more positive value observed for the sintering process ($-14.5 \pm 3.2$‰) and the pelletizing process ($-6.4 \pm 2.5$‰) (Figure S5). This suggests that the mechanisms by which these processes emit NOx were complex and highly variable. The maximum temperature in the central area of the hot air stove can exceed 2000 °C, with the majority of emitted NOx being thermal NOx (Toof, 1986). Due to the continuous $^{14}N^{14}N$ supplementation, generated NOx exhibits a negative $\delta^{15}N$. In contrast, the temperatures of sintering and pelletizing processes are relatively low (1200 ~ 1400 °C), with the majority of emitted NOx being fuel-type NOx (Toof, 1986). Specifically, functional groups such as pyrrole and pyridine in coke powder decompose at high temperatures and react with $O_2$ to produce $NO_x$, resulting in a positive value of $\delta^{15}N$-NOx (Hayhurst and Vince, 1980). It should be noted that the raw materials used for sintering were iron ore fines and coke powder, which differed from those used in coal combustion in power plants. Consequently, the $\delta^{15}N$ values of NOx released cannot be considered to be the same source isotopic fingerprint, as they are influenced by differences in $^{15}N$ abundance (Heaton, 1990).

This study also characterizes $\delta^{15}N$-NOx values released from biomass burning (+1.2 ± 3.0 ‰), obtained by burning various types of plant materials locally in Tianjin. The mean values observed in this study were comparable to those reported by Fibiger and Hastings (2016) (+1.0 ± 4.1 ‰), while the range of fluctuations (−4.9 ‰ ~ +5.5 ‰) was narrower than those observed by their results (−7 ‰ ~ +12 ‰) (Figure S4d). Moreover, the $\delta^{15}N$ values of biomass fuel combustion from the Zhejiang Province, China, which measured the $\delta^{15}N$-NOx values of biomass burning from various types of biomass fuels [Shi et al., 2022], were found to be lower than those observed in the present study. These results indicate that the variations in the $\delta^{15}N$-NOx values of biomass burning are primarily influenced by the type of biomass fuel (Shi et al., 2022). Similarly, significant differences were observed between various





types of biomass fuel combustion, supporting this viewpoint (Figure S6). These
findings indicate that the NOx from biomass burning worldwide may exhibit a wide
range of $\delta^{15}$N values due to varying $\delta^{15}$N-biomass values. Consequently, in order to
accurately assess the contribution of biomass burning to atmospheric NOx in Tianjin,
it is essential to obtain the $^{15}$N signal of NOx release from typical local biomass
combustion is essential.
In comparison to the five sources mentioned earlier, soil emission exhibits a
lower concentration of NOx, accompanied by the most negative $\delta^{15}$N value (-33.7 ±
9.7 ‰) (Figure 1). Moreover, the concentration or $\delta^{15}$N values of NOx released from
various types of soil demonstrate a clear distinction (Figure S7). The concentration of
NOx released from wheatland soil after irrigation was found to be significantly higher
than that observed prior to irrigation (Figure S7). However, the difference in $\delta^{15}$N of
NOx (-38.5 ± 2.5 ‰ vs. -40.1 ± 5.9 ‰) was not found to be significant (p > 0.05).
These results indicate that irrigation has a significantly impact on NOx release from
wheatland soils, but not on its $\delta^{15}$N value. The $\delta^{15}$N-NOx released from wetland soils
was comparable to that released from wheatland soils, both of which exhibited a more
negative value than those released from urban green belt soils (Figure S7). Previous
studies have indicated that soil NO release is primarily divided into biotic and abiotic
processes (Hall et al., 1996). The biotic process encompasses the nitrification of $NH_4^+$
and the denitrification of $NO_3^-$, while the abiotic process is primarily the chemical
reduction of $NO_2^-$ in soil (Yu and Elliott, 2017). Kinetic processes favor the presence
of $^{14}$N in the product gases derived from the biotic process, whereas the $\delta^{15}$N-NOx
released by abiotic processes in soil is significantly positive than that of biotic
processes (Felix and Elliott, 2014;Li and Wang, 2008;Baggs, 2008). Consequently,
our study suggests that a significant abiotic process may have caused NO release from
urban green belt soils. Given the abundance of wheatland in the vicinity of Tianjin
and the urban area covered by green belts and coastal wetlands, we used the
aforementioned $\delta^{15}$N-NOx values from the three soil types as representative of local
soil emissions in Tianjin.
3.2 Characteristics of concentration and $\delta^{15}$N value of $NO_3^-$ in $PM_{2.5}$
3.2.1 Concentrations of $PM_{2.5}$ and $NO_3^-$
The $PM_{2.5}$ concentration ranged from 5.1 μg m$^{-3}$ to 297.8 μg m$^{-3}$ during the
sampling period in Tianjin, with an average value of 68.6 ± 62.4 μg m$^{-3}$ (Table 2 and
Figure 2). Pre-heating exhibited the highest mean concentration (75.3 ± 53.3 μg m$^{-3}$),
followed by mid-heating (68.9 ± 43.2 μg m$^{-3}$) and late-heating (39.0 ± 27.9 μg m$^{-3}$).
SNA (Sulfate: $SO_4^{2-}$, Nitrate: $NO_3^-$, Ammonium: $NH_4^+$) constituted the major ions in
$PM_{2.5}$, contributing to over 40% of $PM_{2.5}$ (Figure S8). The concentration of $NO_3^-$
showed a significant positive linear correlation with $PM_{2.5}$ (Figure 2), suggesting that
the substantial increase in $PM_{2.5}$ pollution is linked to an increase in $NO_3^-$
concentration.
The variation pattern of $NO_3^-$ concentration during different sampling periods
aligned with that of $PM_{2.5}$ (Figure 2). The highest concentration of $NO_3^-$ was





observed during pre-heating ($16.0 \pm 12.4$ µg m$^{-3}$), and the lowest concentration was
observed during late-heating ($9.7 \pm 8.7$ µg m$^{-3}$) (Table 2). Notably, $NO_2$, as the
precursor of $NO_3^-$, did not follow the observed pattern of change in $NO_3^-$
concentration. The highest concentration of $NO_2$ was observed during mid-heating
(Table S1 and Figure S7), potentially influenced by increased coal combustion for
heating (Luo et al., 2019). Generally, $NO_2$ concentration and its secondary conversion
efficiency were the key factors affecting the concentration of $NO_3^-$. Therefore, the
difference could be attributed to biases in the secondary conversion efficiency of $NO_2$
(Xiao et al., 2023). This is supported by the lower relative humidity (RH) during mid-
heating ($31.1 \pm 20.3\%$) compared to pre-heating ($46.2 \pm 14.8\%$) (Table S1), as higher
RH could lead to increased $NO_3^-$ formation (Gao et al., 2020). Compared to $NO_3^-$, the
$SO_4^{2-}$ concentration was highest during mid-heating. In addition, $SO_2$, mainly
originating from coal combustion, exhibited a similar variation pattern to $SO_4^{2-}$,
potentially attributed to increased coal combustion for heating (Figure S9) (Feng et al.,
2020). Markers primarily originating from coal combustion, including CO and Cl$^-$
(Figure S9), also showed higher concentration during mid-heating, supporting our
speculation (Luo et al., 2019;Xiao et al., 2022). While the increase in $NO_2$ could be
attributed to enhanced biomass burning during mid-heating, the concentration of $K^+$,
primarily a biomass burning marker (Xiao et al., 2024a), exhibited slight variation
during the three periods and was significantly lower than Cl$^-$ (Table 2). Thus, the
impact of coal combustion heating on $NO_3^-$ sources was evident, despite mid-heating
periods being unfavorable for $NO_3^-$ generation.
3.2.2 Characteristics in $\delta^{15}N$ value of $NO_3^-$
The $\delta^{15}N$-$NO_3^-$ values in this study ranged from -0.7‰ to 20.8‰, with a mean
$\delta^{15}N$ value of $8.5 \pm 4.4$‰ (Figure 2 and Table 2). This measurement was more
negative than the observations reported by Feng et al. (2020) ($14.1 \pm 3.2$‰) in Tianjin
in 2017, Luo et al. (2019) in Beijing ($13.9 \pm 2.4$‰) in 2013, and Zhang et al. (2021) in
Beijing ($+11.5 \pm 5.0$‰) in 2018. The Coal Replacement Project, initiated since 2017
to replace coal with cleaner energy sources such as natural gas and electricity in major
cities such as Beijing and Tianjin in northern China (Feng et al., 2020), could explain
this reduction in coal use leading to a gradual decrease in $\delta^{15}N$ values of $NO_3^-$, as
NOx emissions from coal combustion have positive $\delta^{15}N$ values. This speculation is
supported by reported changes in the Bohai Sea from 2014 to 2019 (Zong et al.,
2022b).
Significant differences were observed in $\delta^{15}N$-$NO_3^-$ values among the three
sampling periods in this study. The most positive value was observed during mid-
heating ($12.4 \pm 3.3$‰), followed by pre-heating ($7.7 \pm 4.1$‰) and late-heating ($7.1 \pm$
$4.1$‰) (Table 2). These results suggest variations in the sources of NOx during three
sampling periods. For instance, the primary source $\delta^{15}N$-NOx from coal combustion
($+12.3 \pm 1.7$ ‰) has most positive value (Figure 1), indicating coal combustion as the
dominant source of NOx during mid-heating periods (Luo et al., 2019). However, for
the other sampling periods, the $\delta^{15}N$ values of $NO_3^-$ indicate multiple sources of NOx,
including fossil fuel combustion, industrial emissions, biomass burning, and soil





sources (Zong et al., 2017;Sun et al., 2020). Previous studies conducted in Chinese
cities have reported monthly variations in $\delta^{15}N$-$NO_3^-$, which could also be attributed
to changing NOx sources (Luo et al., 2020a;Luo et al., 2020b;Guo et al., 2021;Zhang
et al., 2022).
**3.3 The importance of $\delta^{15}N$ values from local NOx source for sources**
**apportionment of $NO_3^-$ in $PM_{2.5}$**
In this study, the MixSIAR model was employed to quantify the contribution of
NOx sources. Previous studies have estimated the contribution of NOx sources using
the MixSIAR model, based on the known $\delta^{15}N$ values of NOx from different sources.
These studies have focused on four sources of NOx: coal combustion, biomass
combustion, vehicle exhaust, and soil sources (Zong et al., 2017;Zong et al.,
2020b;Zhao et al., 2020;Zhang et al., 2020). To facilitate comparison, the $\delta^{15}N$ data of
above four sources from previous studies (Scenario 1) and this study (Scenario 2)
were input into the MixSIAR model to quantify the sources of $NO_3^-$, respectively
(Table 1). Throughout the entire sampling duration, the average contributions
estimated by the MixSIAR model exhibited no substantial disparities between
Scenarios 1 and 2, suggesting that localized $\delta^{15}N$ data acquisition for NOx sources
might be superfluous. However, the contributions of individual NOx sources to $NO_3^-$
in $PM_{2.5}$ were found to be significantly different during a certain sampling period
when calculated using different $\delta^{15}N$ data for NOx sources. For instance, during the
pre-heating periods, the contributions of soil sources, coal combustion, and biomass
burning in Scenario 1 were $23.0 \pm 10.1\%$, $17.8 \pm 12.5\%$, and $24.1 \pm 17.3\%$,
respectively. The contributions were slightly lower than the estimated results in
Scenario 2 ($25.0 \pm 7.4\%$, $19.1 \pm 13.2\%$, and $25.6 \pm 18.3\%$). Furthermore, the
contribution of vehicle exhaust exhibited a notable discrepancy, being considerably
higher in Scenario 1 ($35.1 \pm 22.8\%$) compared to Scenario 2 ($30.2 \pm 21.1\%$).
Therefore, the calculation of the contributions of various sources to $NO_3^-$ using $\delta^{15}N$
data in NOx sources from previous studies may result in inaccuracies.
The uncertainty index ($UI_{90}$), derived from posterior distribution data (as detailed
in Text S3), serves as a metric to evaluate the uncertainty in the results calculated by
the MixSIAR model (Zhang et al., 2024a). A low $UI_{90}$ value indicates a low degree of
uncertainty, which suggests that the results of the source contribution were stable
(Shang et al., 2020). As shown in Figure 4, the $UI_{90}$ values of coal combustion and
biomass combustion were lower in Scenario 1 than in Scenario 2, indicating that the
results in Scenario 1 were relatively stable. However, the contributions of vehicle
exhaust and soil sources in Scenario 2 were relatively stable, as their $UI_{90}$ values were
lower in Scenario 2 than in Scenario 1. It can therefore be observed that the
uncertainty in contributions from different sources exhibited a variety of degrees of
variability that were influenced by the differing end-member values inputted into the
model. Generally, the correlation of probability density functions (PDFs) between
different sources may provide insight into the validity of model calculations (Parnell
et al., 2010). For instance, if the two sources cannot be completely differentiated by
the model, their correlation in PDFs will exhibit a strong negative correlation (Lin et





al., 2021). The study revealed a significant negative correlation between the PDFs of vehicle emissions and coal combustion and soil sources both in both Scenario 1 and Scenario 2 (Table S2), indicating that these sources cannot be completely differentiated. Therefore, the inclusion of additional sources is recommended to enhance the accuracy of estimates provided by the MixSIAR model (Lin et al., 2021).

Since the initiation of the Coal Replacement Project in 2017, the contribution of natural gas combustion to $NO_3^-$ may not be negligible in recent years in Tianjin (Meng et al., 2022;Wang et al., 2022). However, previous studies have seldom examined the role of natural gas combustion in contributing to $NO_3^-$ in $PM_{2.5}$, due to limited availability of reported of $\delta^{15}N$ values of NOx resulting from natural gas combustion (Zong et al., 2022b;Walters et al., 2015b). Consequently, the results may be subject to some degree of uncertainty when only the four $\delta^{15}N$-NOx end-member values are considered. Therefore, we refer to the $\delta^{15}N$-NOx end-member values from natural gas combustion obtained from previous studies (Scenario 3) and locally acquired in Tianjin (Scenario 4) to calculate the relative contribution fractions of the five NOx sources using the MixSIAR model (Figure 4c and 4d).

In contrast to the findings of the four sources (Scenario 1 and Scenario 2), significant discrepancies exist between Scenario 3 and Scenario 4. Especially the contribution fractions of natural gas combustion (21.0 ± 13.8% *vs*. 16.5 ± 11.5%) and coal combustion (18.2 ± 10.7% *vs*. 22.0 ± 12.7%), the results estimated in Scenario 4 significantly differ from those in Scenario 3. These disparities are also present across different sampling periods. During pre-heating periods, contributions of vehicle exhaust (24.9 ± 18.5% *vs*. 25.6 ± 19.0%) and biomass burning (20.9 ± 15.1% *vs*. 24.1 ± 17.2%) were lower in Scenario 4 compared to Scenario 3. Conversely, natural gas combustion (21.5 ± 14.3% *vs*. 17.3 ± 11.3%) and soil sources (14.4 ± 9.7% *vs*. 13.2 ± 8.7%) estimates in Scenario 3 were higher than those in Scenario 4. Similar differences were observed during the mid-heating periods. However, in the late-heating periods, contributions of vehicle exhaust (22.1 ± 18.0% *vs*. 26.0 ± 17.3%) and coal combustion (15.6 ± 10.8% *vs*. 18.2 ± 11.6%) calculated in Scenario 4 was higher than those in Scenario 3. In addition, biomass burning (20.8 ± 14.9% *vs*. 20.6 ± 14.9%), natural gas combustion (24.1 ± 16.1% *vs*. 18.7 ± 13%) and soil sources (17.4 ± 10.8% *vs*. 16.4 ± 9.9%) in Scenario 4 were lower than those in Scenario 3. In both scenarios, the contribution of natural gas combustion to $NO_3^-$ was close to or even exceeds that of soil sources (Figure 4). This underscores the need to consider natural gas combustion when assessing $NO_3^-$ sources in $PM_{2.5}$, particularly in urban areas impacted by the Coal Replacement Project (Zhang et al., 2024a). Consequently, our result further highlight that the natural gas combustion as a source input the model could improve the validity of the calculations to a certain extent. Additionally, measuring the $\delta^{15}N$ values of the local NOx source is necessary to accurately identify the source of $NO_3^-$ in $PM_{2.5}$.

### 3.4 Industrial emission should be an important source of $NO_3^-$

Industrial emissions, particularly those from the iron and steel sector, consume a



significant quantity of fossil fuel and mineral resources, resulting in a notable increase in NOx emissions (Wang et al., 2019). The iron and steel industry in China has undergone considerable expansion, resulting in a marked increase in NOx emissions. From 2005 to 2015, emissions escalated from 687.93 kt to 1017.24 kt (Gao et al., 2019). This trend suggests that emissions from this sector increasingly affect urban atmospheric NOx levels, especially in industrial cities. Our investigation has revealed that the $\delta^{15}$N-NOx signature from the iron and steel industry is distinct from that of other sources, such as vehicle exhaust, coal combustion, and natural gas combustion (Figure 1). Consequently, it is necessary to treat this source as a unique end-member in the apportionment of $NO_3^-$.

The MixSIAR model was used to estimate the contributions of six NOx sources (coal combustion, biomass burning, vehicle exhaust, soil sources, natural gas combustion and industrial emission source) to $NO_3^-$ in $PM_{2.5}$ based on their respective $\delta^{15}$N values. We concluded that coal combustion (22.8 ± 11.9%) was major sources of $NO_3^-$ in $PM_{2.5}$ in Tianjin, followed by the biomass burning (20.9 ± 15.0%) and vehicle exhaust (20.0 ± 17.2%) (Figure S8). In comparison to a previous study in Tianjin (Xiao et al., 2023), where coal combustion contributed 42.6% of NOx, our study observed a significant decrease, attributable to the ultraclean transformation of coal combustion processes. Nevertheless, energy from coal combustion, which remains the main source of NOx in Tianjin, is used for most of the manufacturing and residential sectors. Moreover, there is a possibility that the contribution of coal combustion increased due to long-range transportation of air masses, alongside local emissions (Li et al., 2023). This is particularly significant as there was notable coal combustion activity upwind of Tianjin (Feng et al., 2020;Xiao et al., 2024b). However, the contribution of vehicle exhaust was slightly higher than the results in our earlier study (19.8%) (Xiao et al., 2023), which resulted from the increased of vehicle ownership. It was worth noting that the influence of biomass burning was also decreased than that in earlier study in Tianjin, which attributed to the effective implementation of measures, such as the ban on straw burning in the North China Plain (Huang et al., 2021). In this study, industrial emission source was accounting for 14%, slightly lower than vehicle exhaust. According to the community emission data system, previous studies have estimated the industrial contribution to NOx to be around 14% (Bekker et al., 2023), which close to our estimate result. Moreover, the contribution of the industrial emission source was found to be greater than that of soil sources (10%) and natural gas combustion (12%), indicating that it should be considered an important source of $NO_3^-$.

Generally, the contribution of certain NOx sources decreased as the number of sources increased (Figure S10). In particular, the contributions from soil and vehicle exhaust sources decreased by 13% and 11%, respectively, compared to the results from four sources. Furthermore, they further decreased by 4% and 4%, respectively, compared to the results from five sources. In this case, however, the contribution of coal combustion was slightly increased (Figure S10). The could indication that the number of sources will markedly influence the estimate results by the MixSIAR



model. Furthermore, the correlations of PDFs between coal combustion and biomass
burning remained unchanged, while those between the other sources decreased further
when the number of sources input into the model increased from four to six (Table
S2). This indicated that the inter-influence between these sources was further reduced,
and the model was able to distinguish between them (Lin et al., 2021). Moreover, the
contributions of all sources demonstrated more relatively stable results, with $UI_{90}$
values exhibiting the lowest values compared to the results estimated by the four or
five sources (Figure 4c) (Zhang et al., 2024a). This is because after setting the total
contribution of all sources in the model to 1, the lack of input sources in the model
may lead to an increase in the fluctuation of the calculated results (Lin et al.,
2021;Zhang et al., 2024a;Feng et al., 2023). Therefore, it was concluded that
incorporating industrial sources in MixSIAR model could decrease uncertainty in
calculating the contribution of NOx sources.

To further elucidate the reasonable of $NO_3^-$ source apportionment results when
the $\delta^{15}N$ signature of NOx from industrial emission sources were input the MixSIAR
model, we examine the simulation results of different sampling periods, as shown in
Figure 5. It was demonstrated that, irrespective of the sampling period, the
contribution of NOx sources varied with the number of sources increased, attributable
to the sensitive of MixSIAR model to missing emission sources (Feng et al., 2023).
Notably, during the mid-heating periods, coal combustion and biomass burning
contributed more significantly than in other periods (Figure 6), collectively exceeding
50% when six sources were incorporated into the MixSIAR model. These findings
suggest that winter heating emissions play a dominant role in the increase of $NO_3^-$
concentrations in the urban area of Tianjin (Luo et al., 2019;Zhao et al., 2020).
Additionally, the contributions of these two sources were higher during pre-heating
compared to late-heating periods, when six sources were considered in the MixSIAR
model. These differences were also reflected in the concentrations of trace factor from
biomass burning ($K^+$) and coal combustion ($Cl^-$) (Sun et al., 2020;Zong et al., 2018)
(Figure 6c and 6d). However, the simulation results did not exhibit similar
consistency when five or four sources were included in the MixSIAR model. For
instance, the contribution of coal combustion was very close in the pre-heating and
late-heating periods when five or four sources were considered. Although the
contribution of soil sources exhibited similar varied patterns across the three sampling
periods when different number of emission sources were considered in the MixSIAR
model (Figure 5), the contribution of soil sources estimated with six sources was
significantly lower than the results from five or four sources. Especially in the mid-
heating period, its contribution was less than 10%. This was expected as low soil
temperatures decrease NOx emissions into the atmosphere (Lin et al., 2021).
Interestingly, vehicle exhaust was the highest contributor to NOx during the late-
heating periods (Figure 5), mainly attributable to the gradual weakening of coal
combustion and biomass burning activities for heating as temperatures increase.
Although vehicle exhaust emissions of NOx may increase due to the rapid rise in car
ownership, their contribution fraction was only around 20%, owing to the three-way
catalyst (TWC) and Selective Catalytic Reduction (SCR) equipment installed in petrol





and diesel vehicles to mitigate NOx emissions throughout China (Guan et al.,
2014;Gu et al., 2022). It should be noted that the contribution of industrial sources
(16.2 ± 12.5%) during the late-heating periods was close to coal combustion (17.7 ±
11.2%), further underscoring the importance of incorporating industrial sources in
calculating results using the MixSIAR model. In essence, the exclusion of industrial
sources may lead to an increase of more than 15% in the contribution fraction of other
sources, biasing source contribution estimates and misguiding emissions reduction
measures.
### 3.5 Limitations and outlook
The dataset presented in this study represents, to the best of our knowledge, the
first more systematic attempt to determine the $\delta^{15}$N values of several significant NOx
sources within urban environments in China. However, it is essential to acknowledge
that some sources, like bulk coal combustion, the metallurgical industry, and
residential gas, were either incompletely sampled or not sampled at all. Additionally,
the $\delta^{15}$N values of NOx emissions from soil across different seasons remain unknown.
These omissions could influence the outcomes of source apportionment, and result in
several uncertainties. Nevertheless, it can be determined that in the calculation results
of the MixSIAR model, the role of local $\delta^{15}$N-NOx source values is critical and
should not be overlooked. And as we introduced more sources into the model, the
estimates of the contribution of each NOx source grew steadier, and the mutual
influence among these sources diminished significantly. This also highlights the
importance of comprehensively determining the $\delta^{15}$N values of typical NOx sources.
Therefore, it would be beneficial for $NO_3^-$ source apportionment to further refine the
NOx source types and improve the $\delta^{15}$N values of other NOx sources in the future.
It is widely recognized that different conversion pathways for NOx to $NO_3^-$
exhibit clear isotopic fractionation of nitrogen. This can lead to inaccuracies in
estimating the contributions of nitrate sources, particularly because the specific
influence of various pathways on the fractionation coefficient εN (NOx → $NO_3^-$)
often remains indistinct (Feng et al., 2020;Zhang et al., 2019). Specialized pathways,
such as those involving heterogeneous chlorine chemistry and nitrogen trioxide, can
alter the $\delta^{15}$N values of $NO_3^-$ (Luo et al., 2023;Zhang et al., 2024b). In this study, the
$\delta^{18}$O-$NO_3^-$ values helped constrain the fractionation factor from NOx to $NO_3^-$ (Xiao
et al., 2020), but only two primary pathways, hydroxyl radical oxidation and nitrogen
pentoxide hydrolysis, were taken into account. Previous research supports the view
that these pathways account for up to 95% of $NO_3^-$ production (Lin et al., 2021;Xiao
et al., 2020), implying that alternative pathways might exert a relatively minor impact
on εN calculations. Nonetheless, future measurements of $\Delta^{17}$O- $NO_3^-$ are essential to
elucidate the isotopic fractionation coefficients comprehensively during the formation
of $NO_3^-$.
## 4. Conclusions
In this study, the $\delta^{15}$N values of 6 NOx sources in the local Tianjin area collected



by the active sampler were determined. Results shown that $\delta^{15}N$ value of NOx emissions from coal combustion exhibited the positive value (+12.3 ± 1.7‰), followed by the biomass burning (+1.2 ± 3.0‰), the vehicle exhaust (-5.2 ± 5.4‰), the industrial emission source (-20.6 ± 16.8‰), the natural gas combustion (-24.8 ± 5.6‰) and the soil sources (-33.7 ± 9.7‰). The observation of significant differences in $\delta^{15}N$-NOx values from disparate sources serves to demonstrate the representative nature of these isotopic fingerprints. Furthermore, the mean values or fluctuation range of $\delta^{15}N$-NOx for almost all sources differed from the values reported abroad in previous studies, suggesting that the $\delta^{15}N$ values of NOx sources has local characteristics.

The contributions of various NOx sources to $NO_3^-$ in $PM_{2.5}$ during sampling periods were estimated based on the MixSIAR model. In result, coal combustion, biomass burning and vehicle exhaust collectively contributed more than 60%, dominating the sources of $NO_3^-$ during sampling periods in Tianjin. However, the relative contribution fraction of each sources shown clear difference when the $\delta^{15}N$-NOx source data from previous studies and this study inputted into the model, respectively. Coal combustion, in particular, has a relative contribution that may be underestimated without considering the localized characteristics of the isotopic fingerprints of NOx source. Remark, as the number of source inputs in the model increases from four to six, the interpretability of the estimated results for the contribution of each source increases. Moreover, the contribution of various NOx sources was becoming more stable, and the inter-influence between various sources was significantly reduced. Specific examples include the values of $UI_{90}$ and PDFs, both of which exhibited a significant downward trend as the number of sources increased. Overall, the refined $\delta^{15}N$ values of NOx sources have been demonstrated to be an effective tool in distinguishing source contributions of $NO_3^-$, which could help to reduce the uncertainties and inter-influence of each source.

**Acknowledgments**

This work was supported by the National Natural Science Foundation of China (Grant Nos. 42273020 and 41773006), the Postdoctoral Fellowship Program of China Postdoctoral Science Foundation (Grant No. GZC20231552), the Natural Science Foundation of Tianjin (Grant No. 22JCQNJC00700), the Natural Science Foundation of Henan (222300420128).

**Data availability**

The datasets used in this study are available at https://doi.org/10.5281/zenodo.11392166 (Xiao et al., 2024c)

**Conflict of interest.**

The authors declare no conflicts of interest relevant to this study.



**Author contributions.**

Hao Xiao, Qinkai Li and Xiaodong Li designed the study. Hao Xiao, Qinkai Li, Wenjing Dai and Gaoyang Cui performed field measurements and sample collection; Hao Xiao and Qinkai Li performed chemical analysis; Hao Xiao and Qinkai Li performed data analysis; Hao Xiao wrote the original manuscript; and Shiyuan Ding, and Xiaodong Li reviewed and edited the manuscript.

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



Table 1 Comparison of the $\delta^{15}N$ characteristic spectra of $NO_x$ sources reported in the previous and
present study

| NO_x sources | Previous studies | | | | This study | | |
|---|---|---|---|---|---|---|---|
| | av (‰) | std (‰) | n | Reference | av (‰) | std (‰) | n |
| Coal combustion | +13.7 | 4.6 | 47 | (Walters et al., 2015b;Felix et al., 2012) | +12.3 | 1.7 | 13 |
| Vehicle exhaust | −7.2 | 7.8 | 151 | (Walters et al., 2015b;Walters et al., 2015a;Felix and Elliott, 2014;Heaton, 1990) | −5.2 | 5.4 | 62 |
| Biomass burning | +1.0 | 4.1 | 24 | (Fibiger and Hastings, 2016;Felix and Elliott, 2013;Hastings et al., 2009) | +1.2 | 3.0 | 14 |
| Soil emission | −33.8 | 12.2 | 6 | (Felix and Elliott, 2014;Li and Wang, 2008) | −33.7 | 9.7 | 12 |
| Nature gas combustion | −16.5 | 1.7 | 23 | (Walters et al., 2015b) | −24.8 | 5.6 | 5 |
| Industrial source | N/A | N/A | | N/A | −20.6 | 16.8 | 17 |

Note:N/A represents data unknown.





Table 2 Mean mass concentrations of $PM_{2.5}$ and water-soluble inorganic ions as well as the $\delta^{15}N$-
$NO_3^-$ in Tianjin urban areas at different sampling stages for 2018 ~ 2019 (av ± std)

| Species | All days Average (n = 142) | Pre-heating Average (n = 56) | Mid-heating Average (n = 30) | Late-heating Average (n = 56) |
|---|---|---|---|---|
| $PM_{2.5}$ (µg m$^{-3}$) | 68.6 ± 62.4 | 75.3 ± 53.3 | 68.9 ± 43.2 | 39.0 ± 27.9 |
| $SO_4^{2-}$ (µg m$^{-3}$) | 4.9 ± 4.2 | 5.9 ± 4.2 | 6.1 ± 6.0 | 3.1 ± 1.9 |
| $NO_3^-$ (µg m$^{-3}$) | 12.7 ± 10.8 | 16.0 ± 12.4 | 11.9 ± 9.4 | 9.7 ± 8.7 |
| $NH_4^+$ (µg m$^{-3}$) | 7.7 ± 7.2 | 10.4 ± 9.0 | 7.9 ± 6.0 | 4.8 ± 4.0 |
| $Cl^-$ (µg m$^{-3}$) | 2.3 ± 2.1 | 2.7 ± 2.0 | 4.0 ± 2.5 | 0.9 ± 8.4 |
| $K^+$ (µg m$^{-3}$) | 0.7 ± 0.6 | 0.8 ± 0.7 | 0.9 ± 0.5 | 0.4 ± 0.3 |
| $Ca^{2+}$ (µg m$^{-3}$) | 0.2 ± 0.1 | 0.1 ± 0.1 | 0.1 ± 0.1 | 0.2 ± 0.1 |
| $Na^+$ (µg m$^{-3}$) | 0.2 ± 0.1 | 0.2 ± 0.1 | 0.3 ± 0.1 | 0.2 ± 0.1 |
| $Mg^{2+}$ (µg m$^{-3}$) | 0.01 ± 0.03 | 0.03 ± 0.02 | 0.03 ± 0.01 | 0.03 ± 0.02 |
| $\delta^{15}N$ (‰) | 8.5 ± 4.4 | 7.7 ± 4.1 | 12.4 ± 3.3 | 7.1 ± 4.1 |

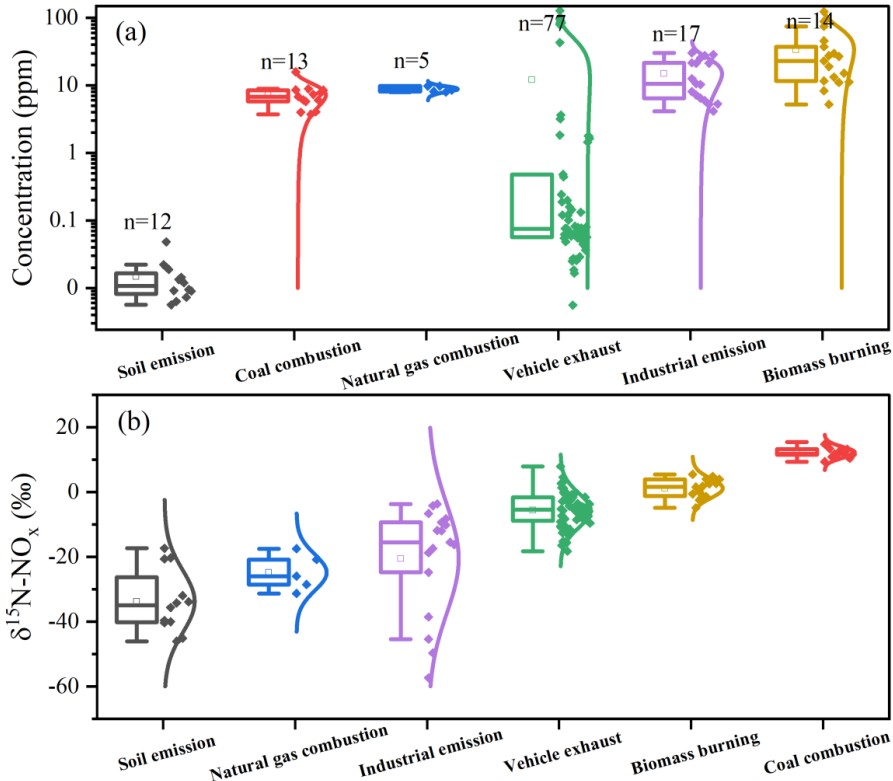

Figure 1 Concentration (a) and (b) $\delta^{15}N$ value of NOx in each emission sources. The box-whisker
plot symbols represent the 25th–75th percentiles. The curved lines to the right of the box-whisker
plot symbols illustrate the probability distribution of the sample points, each of which represents
one sample.



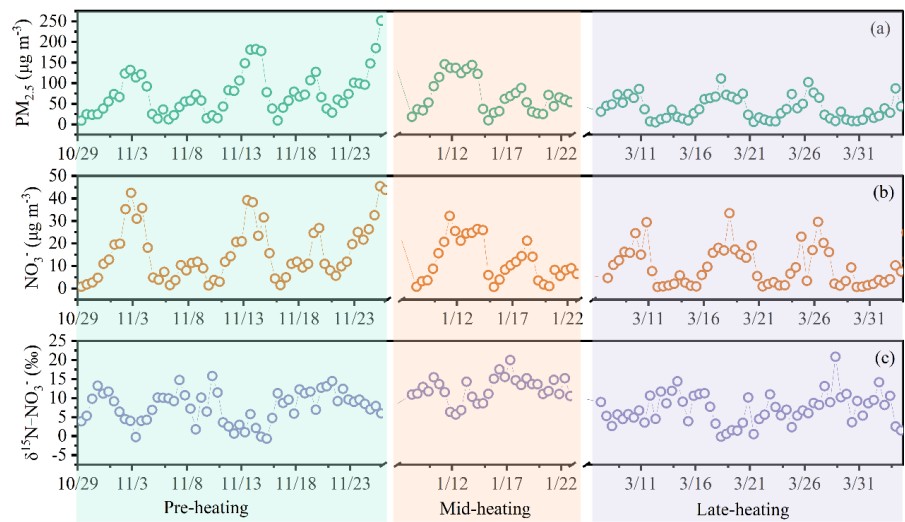


Figure 2 Time series of the concentrations in PM$_{2.5}$ (a) and NO$_3^-$ (b), as well as the δ$^{15}$N values (c)
in NO$_3^-$.



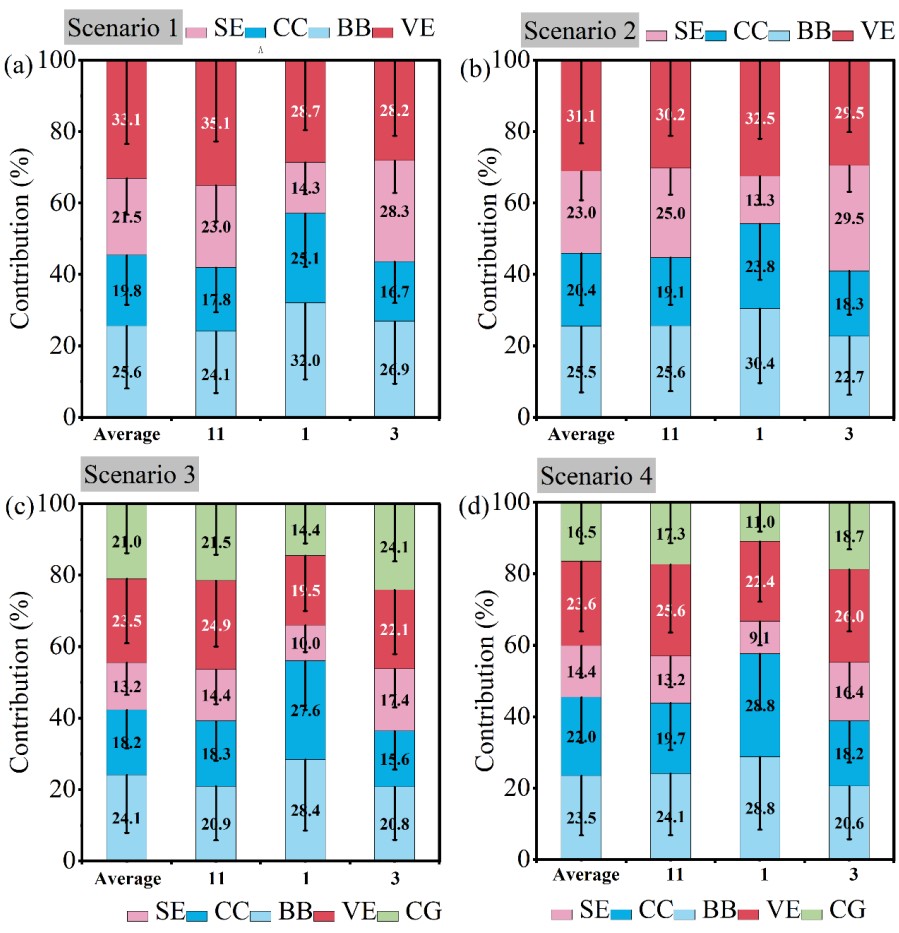

Figure 3 Comparison of fractional contributions of $NO_3^-$ sources in $PM_{2.5}$ in Tianjin estimated by different $\delta^{15}N$ values of NOx sources. The results of Scenario 1 and Scenario 3 were estimated using the $\delta^{15}N$ values of four and five NOx sources obtained from previous studies, while the results of Scenario 1 and Scenario 3 were estimated using the $\delta^{15}N$ values of four and five NOx sources obtained from this study. Also, SE = soil emission, CC = coal combustion, BB = biomass burning, VE = vehicle emission, and CG = combustion of natural gas.



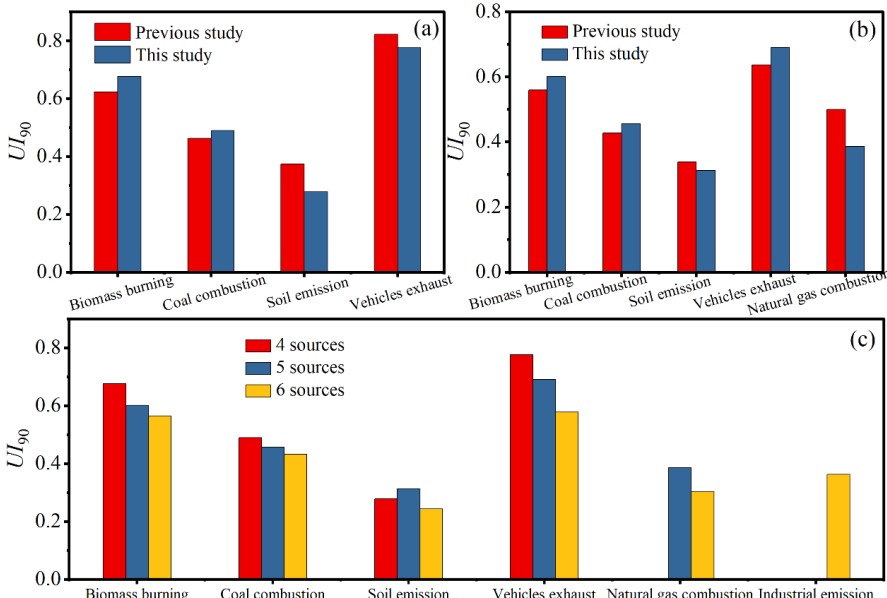

Figure 4 (a) The $UI_{90}$ values of the contribution fraction of the four sources estimated by the isotopic fingerprint in NOx sources obtained from previous studies and this study. (b) The $UI_{90}$ values of the contribution fraction of the five sources estimated by the isotopic fingerprint in NOx sources obtained from previous studies and this study. (c) The $UI_{90}$ values of the contribution fraction of the four, five, and six sources estimated by the isotopic fingerprint in NOx sources obtained from this study.



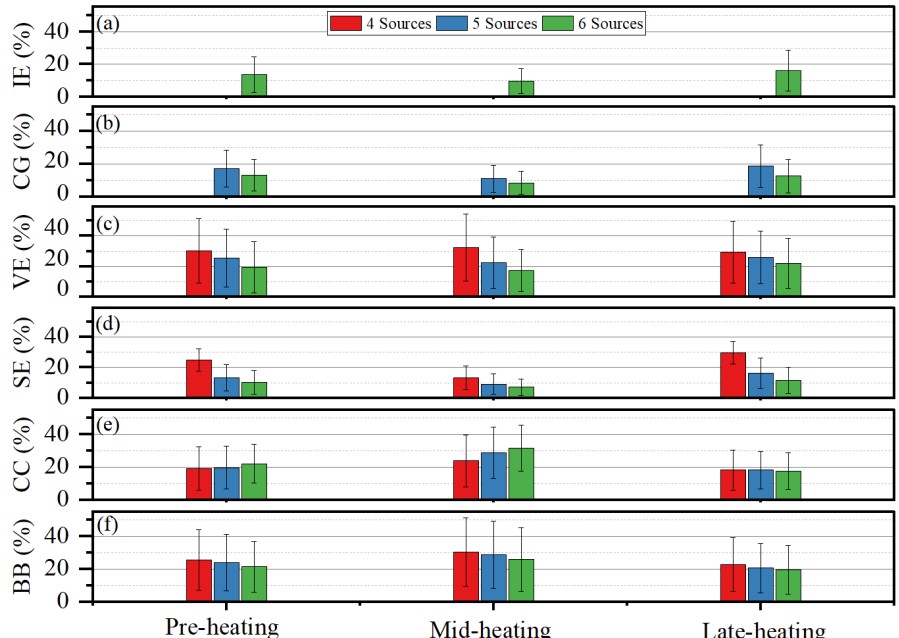

901

Figure 5 The contribution fraction of the four, five, and six sources in different periods estimated
by the isotopic fingerprint in NOx sources obtained from this study. Also, SE = soil emission, CC
= coal combustion, BB = biomass burning, VE = vehicle emission, and CG = combustion of
natural gas.



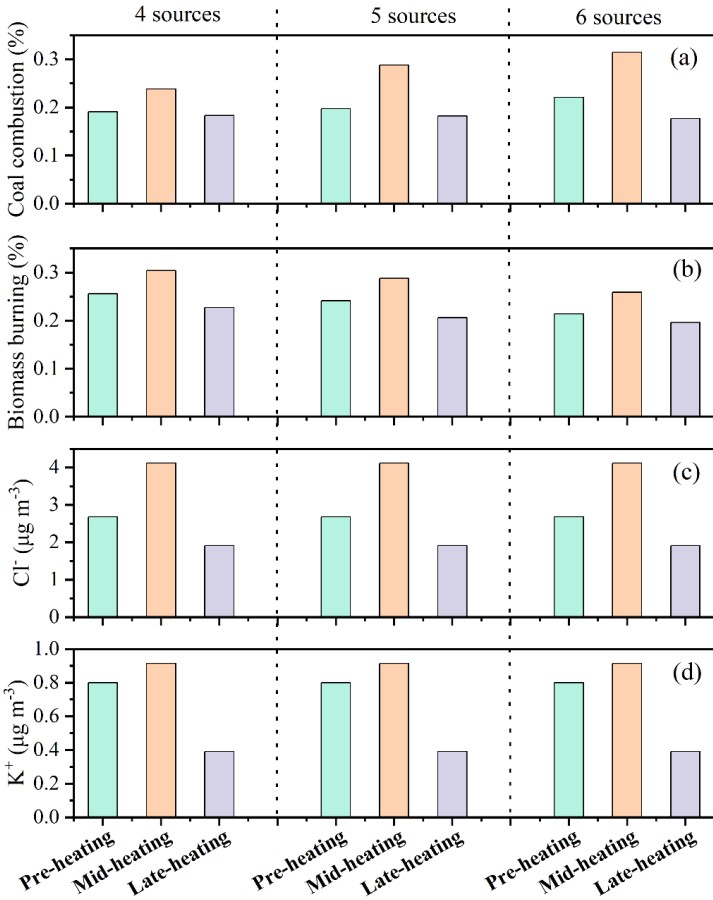

Figure 6 The version trend in contribution fractions of coal combustion (a) and biomass burning (b) in different periods estimated by the isotopic fingerprint in NOx sources of four, five and six sources obtained from this study. (c) and (d) were the version trend in concentrations of Cl⁻ and K⁺ during different sampling periods.



**Graphical Abstract**

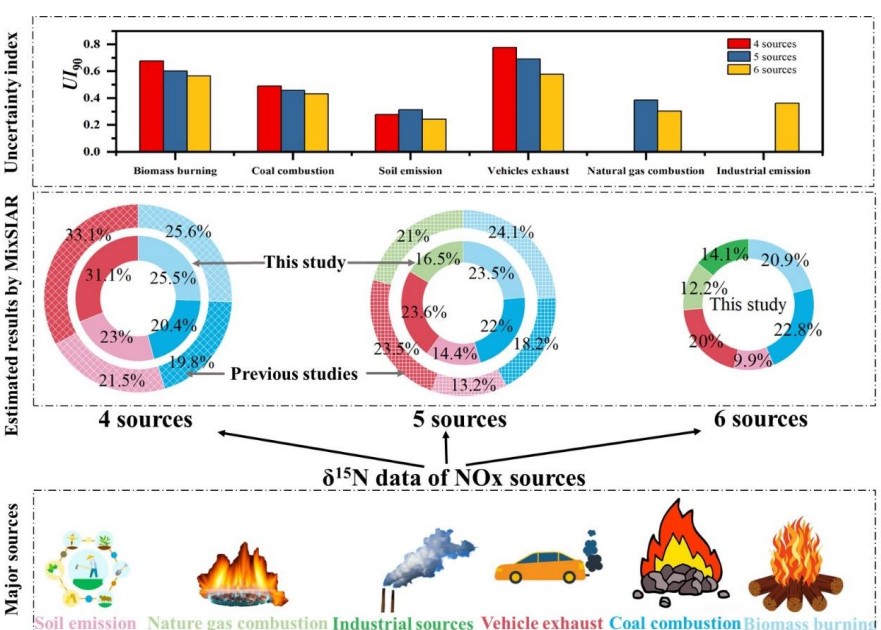


