# Peer review of "Technical note: Refining $\delta^{15}$ N isotopic fingerprints of local"

_EGUsphere, 2024_

## Author Comment (AC1)

**Detailed response to the reviewers' comments**

Atmospheric Chemistry and Physics (Paper egusphere-2024-1621)

Title: "Technical note: Refining $\delta^{15}$N isotopic fingerprints of local NO$_x$ for accurate source identification of nitrate in PM$_{2.5}$"

Dear Editor(s) and reviewers,

We would like to express our gratitude for your valuable comments and suggestions. The manuscript has been carefully revised according to the comments.

Response to the editor(s)' comments and detailed point-by-point responses to the reviewers' comments are provided in the following pages. Please note that the comments are presented in *italics font*, and our responses are in Roman font and marked in blue. In addition, all the line numbers in the responses refer to the revised manuscript. All changes made to the manuscript are marked in red.

Please let us know if you have any further questions regarding our manuscript.

We are looking forward to your reply.

Sincerely,

Pro. Xiaodong Li

E-mail: xiaodong.li@tju.edu.cn

Institute of Surface-Earth System Science, School of Earth System Science

Tianjin University, Tianjin, P. R. China, 300072

**Reviewer #1:**

*The authors present a manuscript detailing various stable nitrogen isotope ratio measurements of major nitrogen oxide ($NO_x$) sources and use these values for particulate nitrate source apportionment during a typically heavily polluted period in Tianjin, China. This manuscript provides an impressive wealth of new $\delta^{15}N(NO_x)$ emission source measurements. Specifically, they have quantified $\delta^{15}N(NO_x)$ values for previously unmeasured $NO_x$ sources, such as industrial emissions, and have made measurements representative of emissions in China, contrasting with most literature values which are predominantly for the US, potentially influencing the results due to different combustion and emission control technologies. They have also employed appropriate measurement techniques (Fibiger et al., 2014), crucial since some previous $\delta^{15}N(NO_x)$ measurements utilized a wide range of $NO_x$ collection techniques that may not all be suitable for $\delta^{15}N(NO_x)$ source characterization. I strongly support their initiative to enhance the $\delta^{15}N$ values of $NO_x$ sources.*

*However, I have major reservations about this work, including its presentation and application for particulate nitrate source apportionment. There appears to be significant confusion in their literature review and citations, which require substantial editing. I have tried to point out some areas that need to be refined, though I suspect that there might be others as well. While the authors are commended for attempting to correct for isotope fractionation, I believe there are issues with their approach. They should have reported the $\delta^{18}O(NO_3^-)$ data and calculated fractionations necessary for accurate source apportionment. This approach to isotope fractionation correction is described in the supplement, but the actual $\delta^{18}O(NO_3^-)$ data is not included, shown, or discussed in the present work. Additionally, it is unclear whether the mixing model results are valid or contribute any meaningful insight into $NO_x$ emission source apportionment. The model is significantly under-constrained (4-5 parameters for one variable), making the output nonsensical. In conclusion, I believe this paper has the potential to be publishable after significant revisions and would recommend resubmission.*

**Response:** We would like to express our sincerest gratitude for your invaluable feedback, which has been instrumental in enhancing the quality of our manuscript. We have endeavored to optimize the manuscript and have incorporated the requisite changes, highlighted in red in the revised paper, which will not impact the content or framework of the paper. We hope that these amendments will meet with your approval. Firstly, we have revised both the literature review and inappropriate citations, including but not limited to errors noted by the reviewers. Second, we have added a discussion of the data on $\delta^{18}O$-$NO_3^-$ to the revised manuscript. In addition, we refer to previous studies and add a description of the isotopic fractionation calculation process, which have been widely used in similar studies. Finally, we also refine the discussions in validity of the quantitative results of the model as well and add additional evidence. It is acknowledged that the present study is not without shortcomings, including the fractionation calculations of isotopes and the constraints of the model. However, every effort has been made to improve these aspects in the revised manuscript. Furthermore, more in-depth studies will be conducted in the future to address these shortcomings. We also summaries these deficiencies in section 3.5 Limitations and outlook

of the manuscript. We would like to reiterate our appreciation for your comments and suggestions. Below, we will provide detailed one-on-one revisions and responses to deficiencies in the original manuscript.

**Major Comments:**

*Question 1: Lines 64-67: Double-check these references. Felix and Elliott, 2014 used a passive sampler and collected $NO_2$ only (not total $NO_x$). Walters et al., 2018 report ambient $\delta^{15}N$ measurements of $NO_2$ and are not direct source emission measurements.*

**Response 1:** Thanks for the comment. This error has been corrected.

Lines 64-70: For instance, Zong et al. (2020a) reported $\delta^{15}N$-$NO_x$ values from vehicle exhaust to be $-8.7 \pm 5.3‰$, using an active sampler with an absorption solution (100 mL, 0.25 mol $L^{-1}$ $KMnO_4$ + 0.50 mol $L^{-1}$ NaOH). However, more negative $\delta^{15}N$-$NO_x$ values (-11.2 ± 6.8‰) were observed for vehicle exhaust collected using a sealed static absorption method, in which 2.8 ml of concentrated sulphury acid ($H_2SO_4$) was mixed with 0.6 ml of 30% hydrogen peroxide ($H_2O_2$) to capture $NO_x$ (Walters et al., 2015a).

*Question 2: Lines 73-76: You might also consider citing, (Fibiger and Hastings, 2016) here.*

**Response 2:** Thanks for the comment. We have added it.

Lines 75-78: However, biomass burns at low temperatures (250 to 1200 °C), and the process produces mainly fuel $NO_x$, with $\delta^{15}N$ depending on the relative abundance of nitrogenous organic matter $^{15}N$ in the biomass itself (Zong et al., 2022a;Fibiger and Hastings, 2016;Jin et al., 2024).

*Question 3: Lines 95-96: Please define "dual isotopic compositions". Do you mean the $^{15}N$ and $^{14}N$ (as dual isotopes of nitrogen), or are you referring to dual isotope deltas of $\delta^{15}N$ and $\delta^{18}O$? If you refer to $\delta^{18}O$, more discussion/explanation is needed here. The $\delta^{18}O$ values are known to derive from the oxidants involved in NOx chemistry and do not reflect NOx sources, which is currently implied by the wording of this sentence.*

**Response 3:** Thanks for the comment. This error has been corrected.

Lines 99-101: Recent research efforts in Tianjin have focused on identifying the sources of $NO_3^-$ through its $\delta^{15}N$ values (Zhang et al., 2019;Xiao et al., 2023).

*Question 4: Lines 132-134: Please indicate the location of the auxiliary pollutant concentrations and meteorology parameters in Fig. S1.*

**Response 4:** Thanks for the comment. We have added it. See in particular the lower right-hand panel in figure S1.

[Figure]

Figure S1 Map of PM$_{2.5}$ and NO$_x$ source sampling locations

**Question 5:** *Line 174: The KnMnO$_4$/NaOH NOx absorbing solution, has been reported to have a large NO$_3^-$ blank (Fibiger et al., 2014). It was mentioned earlier in the text that blanks were evaluated. What were the blank levels, and how were they considered in the reported concentration and isotope data?*

**Response 5:** Thanks for the comment. Results for blank samples and data correction methods have been added in the revised manuscript.

Lines 198-202: A correction was implemented during the data processing stage utilizing the blank samples, with the mean NO$_3^-$ concentration determined to be 6.2 μM and the mean $\delta^{15}$N value established to be 2.8‰ for the blank samples. Thereafter, the $\delta^{15}$N-NO$_x$ was calculated for each sample through a mass balance approach:

$$\delta^{15}\text{N-NO}_x = \frac{\delta^{15}\text{N-[NO}_3^-]_{\text{sample}} - \delta^{15}\text{N-[NO}_3^-]_{\text{blank}}}{[\text{NO}_3^-]_{\text{sample}} - [\text{NO}_3^-]_{\text{blank}}} \qquad (1)$$

**Question 6:** *Lines 177-180: You should cite the original methods paper that describes the denitrifying bacteria method for nitrate isotope analysis (Casciotti et al., 2002; Sigman et al., 2001).*

**Response 6:** Thanks for the comment. We have added it.

Lines 185-187: In this study, we utilized the bacterial denitrification method to determine the dual isotopic values of NO$_3^-$ ($\delta^{15}$N and $\delta^{18}$O) in PM$_{2.5}$ and absorbent solution (Sigman et al., 2001;Casciotti et al., 2002).

**Question 7:** *Lines 183-185: Please cite the papers that report on USGS32, USGS34, and IAEA-N3.*

**Response 7:** Thanks for the comment. We have added it.

Lines 193-196: The study employed three isotopic international standards: USGS32, USGS34 and IAEA-N3, and the analytical accuracies for both $\delta^{15}N$ and $\delta^{18}O$ were ± 0.2‰ and ± 0.3‰, respectively (Böhlke et al., 2003;Sigman et al., 2001).

*Question 8: Lines 202: How is the mixing model output utilized? Most of the mixing model development papers indicate that the average and standard deviation may not be the best metrics for these mixing models, particularly when utilizing one variable ($\delta^{15}N$) to mix between several parameters (4 or 5 sources). Please justify the use of this type of under-constrained model for source apportionment application. For example, if you apply your mixing model to the tunnel measurements (as a test case), would your mixing model results show that vehicles were the dominant source? Or would it suggest many other sources?*

**Response 8:** Thanks for the comment. The Bayesian mixing model is capable of utilizing stable isotope analysis to ascertain the probability distribution of source contributions to an analyzed mixture, while explicitly accounting for the uncertainty associated with the presence of multiple sources, fractionation effects, and isotopic signatures (Moore and Semmens, 2008;Parnell et al., 2010;Parnell et al., 2013). One of the benefits to conducting mixture models in a Bayesian framework is that information from other data sources can be included via informative prior distributions (Moore and Semmens, 2008). Once an informative prior for the proportional contribution of sources is established, MixSIAR can accept the prior as an input during the model specification process.

$$p(f_q \mid data) = \theta(data \mid f_q) \times p(f_q)/\Sigma\theta(data \mid f_q) \times p(f_q)$$

where θ(data|$f_q$) and $p(f_q)$ are the likelihood of the given mixed data and the prior probability of the given state of nature being true based on prior information, respectively. Prior beliefs about proportional source contributions ($f_q$) are defined using the Dirichlet distribution, with an interval of [0, 1] (Zong et al., 2017). Of course, we agree that the average and standard deviation may not be the best metrics for these mixing models, particularly when utilizing one variable ($\delta^{15}N$) to mix between several parameters (4 or 5 sources). However, the average provides a central tendency estimate, giving a clear picture of the most likely contribution from each source, while the standard deviation offers insight into the variability and uncertainty in these estimates. These metrics are widely understood and allow for straightforward comparison between different studies or models (Zong et al., 2022b;Zong et al., 2020b;Walters et al., 2022). Although the model might be under-constrained (given the number of sources relative to the number of isotopic markers), the average and standard deviation help in summarizing the outcomes of the model in a way that is interpretable and useful for decision-making. Furthermore, to enhance the reliability of the study results, the model generated 10,000 potential scenarios for each evaluated potential source (Song et al., 2019;Fan et al., 2020). The more iterations, the more likely that the model output will reflect the true posteriors of the source contributions(Stock et al., 2018). The specific number of iterations required to generate sufficient posterior draws depends on the data, the variances in source isotope signatures and fractionations, and the extent to which the isotope mixture precludes the contribution of specified sources (Moore and Semmens, 2008). A large number of iterations are also important in order to establish an appropriate threshold (T), as the more iterations the model uses to develop a T value, the closer this value will be to the true maximum likelihood of the posterior. If too few iterations are used, the threshold establishment phase of the model run

may yield an inappropriately small T, and this in turn may cause the SIR algorithm to resample a single fq with high likelihood tens or even thousands of times. On the other hand, a strong negative correlation of probability density functions (PDFs) between two different sources indicated that the model was unable to completely differentiate one source from another. The study revealed a significant negative correlation between the PDFs of vehicle emissions and coal combustion and soil sources both in both Scenario 1 and Scenario 2 (Table S2), indicating that these sources cannot be completely differentiated. Therefore, the inclusion of additional sources is recommended to enhance the accuracy of estimates provided by the MixSIAR model (Lin et al., 2021). Moreover, a low $UI_{90}$ value indicates a low degree of uncertainty, which suggests that the results of the source contribution were stable (Shang et al., 2020). In this study, as the number of sources input to the model increased, the contribution of various $NO_x$ sources was becoming more stable (Figure 4c). Taken together, we consider the results of this study to be feasible. Of course, the absence of constraints in the model may introduce some uncertainty into the results of this study. Consequently, further refinement may be necessary in the future to address this issue. It was worth noting that modelling is mainly applied to resolve the ratio of possible contributions from different sources in a mixed and complex environment. The tunnel, as an approximately confined space, has a relatively homogeneous source structure. Therefore, it is not possible to use the change model for source resolution.

Lines 208-222: Briefly, the model initiates with establishing a logistic prior distribution, followed by determining the probability contribution distribution of each source to the mixture. Further details are available in our previous study (Xiao et al., 2023;Li et al., 2021). It was noteworthy that an obvious isotopic fractionation process occurs during the conversion of $NO_x$ to $NO_3^-$. Therefore, the nitrogen isotope fractionation coefficient ($\varepsilon$N) resulting from $NO_x$ to $NO_3^-$ conversion should be calculated prior to determining the relative contribution of $NO_x$ sources using MixSIAR model (Text S2). To enhance the reliability of the study results, the model generated 10,000 potential scenarios for each evaluated potential source (Song et al., 2019;Fan et al., 2020). It was worth noting that the probability density functions (PDFs) of each emission source to $NO_3^-$ should be considered to determine the separation between individual sources (Fan et al., 2020;Lin et al., 2024). Finally, we also have to assess the MixSIAR model calculation result uncertainty based on the uncertainty index ($UI_{90}$) (as detailed in Text S3) of the posterior distribution data (Zhang et al., 2024).

L468-487: The uncertainty index ($UI_{90}$), derived from posterior distribution data, serves as a metric to evaluate the uncertainty in the results calculated by the MixSIAR model (Zhang et al., 2024). A low $UI_{90}$ value indicates a low degree of uncertainty, which suggests that the results of the source contribution were stable (Shang et al., 2020). As shown in Figure 4, the $UI_{90}$ values of coal combustion and biomass combustion were lower in Scenario 1 than in Scenario 2, indicating that the results in Scenario 1 were relatively stable. However, the contributions of vehicle exhaust and soil sources in Scenario 2 were relatively stable, as their $UI_{90}$ values were lower in Scenario 2 than in Scenario 1. It can therefore be observed that the uncertainty in contributions from different sources exhibited a variety of degrees of variability that were influenced by the differing end-member values inputted into the model. Generally, the correlation of PDFs between different sources may provide insight into the validity of model calculations (Parnell et al., 2010). For instance, if the two sources cannot be completely differentiated by the model, their correlation in PDFs will exhibit a strong negative correlation (Lin et al., 2021). The study revealed a significant

negative correlation between the PDFs of vehicle emissions and coal combustion and soil sources both in both Scenario 1 and Scenario 2 (Table S2), indicating that these sources cannot be completely differentiated. Therefore, the inclusion of additional sources is recommended to enhance the accuracy of estimates provided by the MixSIAR model (Lin et al., 2021).

L532-539: Moreover, the correlation of PDFs between any two sources decreased in Scenario 4 than in other Scenario (Table S2). It can be inferred that the inter-influence between these sources diminished in Scenario 4 (Lin et al., 2024). On the other hand, the observed decline in $UI_{90}$ values of all sources when the natural gas combustion input was introduced into the MixSIAR model indicates that the result is a relatively stable calculated outcome (Figure 4) (Zhang et al., 2024). Overall, the performance of the SIAR model for Scenario 4 was much better than other Scenarios.

L593-605: Furthermore, the correlations of PDFs between coal combustion and biomass burning remained unchanged, while those between the other sources decreased further when the number of sources input into the model increased from four to six (Table S2). This indicated that the inter-influence between these sources was further reduced, and the model was able to distinguish between them (Lin et al., 2021). Moreover, the contributions of all sources demonstrated more relatively stable results, with $UI_{90}$ values exhibiting the lowest values compared to the results estimated by the four or five sources (Figure 4c) (Zhang et al., 2024). This is because after setting the total contribution of all sources in the model to 1, the lack of input sources in the model may lead to an increase in the fluctuation of the calculated results (Lin et al., 2021;Zhang et al., 2024;Feng et al., 2023). Therefore, it was concluded that incorporating industrial sources in MixSIAR model could decrease uncertainty in calculating the contribution of $NO_x$ sources.

L664-674: Besides, the absence of constraints in the model may introduce some uncertainty into the results of this study. Consequently, further refinement may be necessary in the future to address this issue. Nevertheless, it can be determined that in the calculation results of the MixSIAR model, the role of local $\delta^{15}$N-$NO_x$ source values is critical and should not be overlooked. And as we introduced more sources into the model, the estimates of the contribution of each $NO_x$ source grew steadier, and the mutual influence among these sources diminished significantly. This also highlights the importance of comprehensively determining the $\delta^{15}$N values of typical $NO_x$ sources. Therefore, it would be beneficial for $NO_3^-$ source apportionment to further refine the $NO_x$ source types and improve the $\delta^{15}$N values of other $NO_x$ sources in the future.

*Question 9: Text S2: This is really important information, and it is somewhat of a shame that it has been moved to the supplement since it is quite important for the source apportionment calculations. One thing to note is that for the nitrogen isotope fractionation for $NO_2$ + OH reaction, the authors only consider the isotope exchange between NO/NO$_2$. However, ambient measurements and models have indicated that this isotope effect is generally quite small and often counterbalanced by NO + O$_3$ reactions (Bekker et al., 2023; Li et al., 2020; Walters et al., 2018), under conditions of high $NO_2$ to $NO_x$ molar ratios. The authors need to justify or revisit the use of this fractionation. Further, what about the isotope effect associated with $NO_2$ + OH. Recent modeling work has suggested that this reaction could be important in setting the $\delta^{15}$N of nitrate (Fang et al., 2021). Once the ε value has been determined (which, again, I have some reservations about), how is it applied to back out the $\delta^{15}$N($NO_x$) data? The authors should have shown the calculated ε values and the backed-out $\delta^{15}$N($NO_x$) results in the main text (note: it is not included*

*in the supplement either). Further, the authors utilize $\delta^{18}O$ data to estimate the nitrate formation pathways. The $\delta^{18}O$ data should be included in the manuscript, as well as the estimated nitrate formation pathways since all of this data is critical in the author's approach for $NO_x$ source apportionment using $\delta^{15}N$.*

**Response 9:** Thanks for the comment. The calculation process for $\varepsilon N$ value of $NO_x$ to $NO_3^-$ has been added in the revised manuscript. Our calculation for the $\varepsilon N$ value of $NO_x$ to $NO_3^-$ has also been widely used in similar studies by previous (Zong et al., 2022b;Zong et al., 2020b;Zong et al., 2017;Fan et al., 2020;Song et al., 2019;Luo et al., 2019;Zhao et al., 2021;Zhao et al., 2020;Li et al., 2023;Li et al., 2022). In this study, only two fractionation processes ($\varepsilon_{\cdot OH}$ and $\varepsilon_{N2O5}$) are considered. We strongly agree with the reviewer's suggestion that the isotope effect associated with $NO_2 + OH$ could be important in setting the $\delta^{15}N$ of $NO_3^-$, especially in under conditions of high $NO_2$ to $NO_x$ molar ratios (Fang et al., 2021;Li et al., 2020). However, we have limited monitoring data to achieve such a precise calculation of the $\varepsilon N$ value of $NO_x$ to $NO_3^-$. Therefore, we quantified the different formation pathway of $NO_3^-$ based on its oxygen isotopes ($\delta^{18}O$) and derived the $\varepsilon N$ value of $NO_x$ to $NO_3^-$. Although previous studies have similarly pointed out that the HC pathway may need to be considered when using $\delta^{18}O\text{-}NO_3^-$ to calculate $NO_3^-$ formation pathways. However, there is a clear positive linear correlation between $\delta^{15}N\text{-}NO_3^-$ and $\delta^{18}O\text{-}NO_3^-$ values ($r = 0.7$, $p < 0.01$), suggesting that $NO_3^-$ is primarily formed through the $\cdot OH$ and $N_2O_5$ pathways (Xiao et al., 2020). Consequently, other pathways of $NO_3^-$ formation, such as $NO_3 + HC$ oxidation pathway, can be disregarded (Walters and Michalski, 2015). More importantly, our study focuses on the importance of local $\delta^{15}N\text{-}NO_x$ source establishment for $NO_3^-$ source resolution. Therefore, we trace the computational methods of our predecessors, both to compare our results with those of our predecessors and to determine the refined isotopic fingerprint in a region-specific context could more effectively distinguish source of $NO_3^-$.

L410-438: As shown in Figure 2c, the $\delta^{18}O\text{-}NO_3^-$ values in this study ranged from 48.3‰ to 102.9‰, with a mean $\delta^{18}O$ value of 81.1 ± 11.5‰ (Table 2). Similar to $\delta^{15}N\text{-}NO_3^-$ values, the most positive $\delta^{18}O\text{-}NO_3^-$ value was observed during mid-heating (89.8 ± 9.9‰), followed by pre-heating (84.5 ± 8.4‰) and late-heating (73.0 ± 9.8‰) (Table 2). Furthermore, a significant positive linear correlation ($r = 0.7$, $p < 0.01$) was identified between $\delta^{15}N\text{-}NO_3^-$ and $\delta^{18}O\text{-}NO_3^-$, indicating that only two predominant oxidation pathways ($\cdot OH$ and $N_2O_5$ hydrolysis) govern $NO_3^-$ formation in this study (Xiao et al., 2020;Walters and Michalski, 2016). Previous studies have shown that the $\delta^{15}N\text{-}NO_3^-$ values in $PM_{2.5}$ does not fully reflect the initial $\delta^{15}N\text{-}NO_x$ due to the fractionation process between $NO_x$ and $NO_3^-$ (Fan et al., 2020;Song et al., 2019). Therefore, we calculated the initial $\delta^{15}N\text{-}NO_x$ values based on the $\delta^{15}N$ and $\delta^{18}O$ values of $NO_3^-$ as follows (Zong et al., 2017). First, the relative contributions of the $\cdot OH$ and $N_2O_5$ pathway were calculated separately using the $\delta^{18}O\text{-}NO_3^-$ values of $PM_{2.5}$. Second, the corresponding $\varepsilon N$ values ($\varepsilon_{\cdot OH}$ and $\varepsilon_{N2O5}$) for the $\cdot OH$ and $N_2O_5$ pathways were estimated by considering the equilibrium isotopic fractionation between $NO_2$ and $NO$, and between $N_2O_5$ and $NO_2$, respectively. Finally, the $\varepsilon N$ value of $NO_x$ to $NO_3^-$ in $PM_{2.5}$ was calculated using the contributions of the two pathways and their corresponding $\varepsilon_{\cdot OH}$ and $\varepsilon_{N2O5}$ values. The detailed procedures for all calculations can be found in the Supporting Information (Text 2). As shown in Figure S11, the contributions of $\cdot OH$ and $N_2O_5$ pathways were 35.4 ± 19.8% and 64.6 ± 19.8%, respectively, suggesting that $N_2O_5$ pathways dominates $NO_3^-$ formation. However, the contributions varied across different sampling

periods, indicating that the pathway for $NO_3^-$ formation also varied. This finding aligns with the results presented in section 3.2.1. The calculated $\varepsilon N$ value of $NO_x$ to $NO_3^-$ were $7.5 \pm 3.4‰$ (Figure S12), close to the results of the previous studies in Beijing (Fan et al., 2020;Song et al., 2019), a large municipality near Tianjin. Furthermore, a slight difference in the $\varepsilon N$ value was found during different sampling periods (Figure S12), further indicating that isotopic fractionation similarly affects the feedback of $\delta^{15}N\text{-}NO_3^-$ to the $NO_x$ source.

***Question 10:*** *Lines 210-214: What about fuel types? There is evidence that gasoline vs diesel-powered vehicles have different $\delta^{15}N(NO_x)$ values (Fibiger et al., 2014; Miller et al., 2017; Walters et al., 2015a, b). Could this have impacted the results if the vehicle-fleet distributions varied between these locations?*

**Response 10:** Thanks for the comment. Fuel types include light petrol National IV, heavy petrol National III and diesel National IV, all of which are commonly used by motor vehicles in the Tianjin area. In addition, $NO_x$ samples were collected at heavily trafficked intersections and tunnels to more accurately characterize the $\delta^{15}N$ signals of $NO_x$ emitted from local vehicle exhausts in Tianjin. Results shown that slight differences were observed between the $\delta^{15}N$ values in vehicle exhaust at different sampling sites (Figure S3). This suggests that our sampling results are representative.

Support Information Text S1: A vehicle exhaust test station in Tianjin was chosen for bench testing (Figure S2b). Fuel types include light petrol National IV, heavy petrol National III and diesel National IV, all of which are commonly used by motor vehicles in the Tianjin area. $NO_x$ emissions from vehicles ($n = 19$) were collected at the tailpipe outlets, meeting the standard with measured $NO_x$ concentrations ranging from 0.1 to 3.6 ppm. Considering the significant range of $NO_x$ emission concentrations, a flow rate of 1 L $min^{-1}$ and a sampling duration of 20 min were established in this study to ensure sufficient $NO_3^-$ for concentration determination and $\delta^{15}N$ analysis. Additionally, $NO_x$ was collected from two specific atmospheric environments: an urban intersection ($n = 18$) (Figure S2b) and a tunnel ($n = 28$) (Figure S2b). The intersection was situated at the junction of Tianjin, with the instrument placed approximately 20 m from the center of the crossroads and at a height of 1.5 m above the ground (Figure S1). Sampling instruments were positioned 30 m from the tunnel entrance ($n = 10$) and at a midpoint 550 m from the exit ($n = 18$), with the extraction ports at a height of approximately 1.5 m above ground level. The intersection was determined to be relatively open and illuminated, with $NO_x$ concentrations ranging from 0.04 to 0.2 ppm during sampling periods. The sampling flow rate was set to 1.5 L $min^{-1}$, requiring approximately 120 min to obtain sufficient nitrogen for isotope measurements. The tunnel space is more confined, with weaker lighting, especially at the midpoint. Wind speed mainly depends on the traveling speed of oncoming vehicles. Monitoring data indicated $NO_x$ concentrations at the entrance and midpoint of the tunnel ranged from 0.006 to 0.08 ppm and 0.03 to 0.1 ppm, respectively. Consequently, the air extraction flow rate was set to 2 to 3 L $min^{-1}$ for approximately 60 minutes.

***Question 11:*** *Lines 214-215: Generally, gasoline has very low nitrogen content, such that "fuel $NO_x$" should be low. Is there evidence that the fuel content utilized in the study location had significant amounts of nitrogen?*

**Response 11:** Thanks for the comment. Gasoline typically has very low nitrogen content, usually below 0.1%. Therefore, the contribution of "fuel $NO_x$" should indeed be minimal in most cases. We have also revised this section.

L242-253: Walters et al. (2015a) observed that fuel NOx exhibited more positive δ15N values than thermal NOx, and catalytic treatment could also yield positive δ15N values. However, the nitrogen content of gasoline is low, typically below 0.1% (Tang et al., 2015), and the contribution of fuel NOx from vehicle exhaust is generally minimal in most cases. When compared with previous studies (Figure S4a and Table 1), our results align with domestic reports but significantly differ from foreign studies (Fisher's least significant difference (LSD) test, the same as below; p < 0.05). The differences in $δ^{15}N$ values of oils, influenced by their generation and depositional settings (Williams et al., 1995), lead to varying $δ^{15}N$ values of fuels NOx. More importantly, the differing emission standards for vehicle exhaust across countries, result in local characteristics of δ15N values of NOx in vehicle exhaust (Zong et al., 2020a).

*Question 12: Lines 222-225: This could only be true if the oil/gas had significant amounts of nitrogen. If the N content is low and $NO_x$ is primarily derived from thermal $NO_x$, then the differences between regions would potentially reflect vehicle fleet differences, combustion differences, and/or combustion technology differences, which would have a direct impact on $δ^{15}N(NO_x)$. It is also important to note here that according to Fig. 1, there is a wide range in emitted $NO_x$ concentration for vehicles (similar to Walters et al., 2015b). In this case, it might be better to utilize a mass-weighted $δ^{15}N(NO_x)$ rather than an unweighted option since the $NO_x$ emission will be weighted towards the heavier $NO_x$ emitters.*

**Response 12:** Thanks for your comment. It is true that petrol is very low in nitrogen and a large amount of motor vehicle exhaust NOx is mainly of the thermal type. Therefore, the differences between regions would potentially reflect vehicle fleet differences, combustion differences, and/or combustion technology differences, which would have a direct impact on $δ^{15}N$-$NO_x$. For instance, Zong et al. (2020a) has reported that the highest $δ^{15}N$-$NO_x$ values displayed a rising trend as emissions standards were continuously updated. Therefore, we have re-analyzed the reasons for the regional differences in $δ^{15}N$-$NO_x$ values in vehicle exhaust in the revised manuscript. In addition, since the NOx concentrations of the vehicle exhaust samples showed great fluctuations, the reviewers suggested that we use a weighted average to express the $δ^{15}N$-$NO_x$ values of vehicle exhaust. We are sorry that we didn't go into explaining the reasons for the large fluctuations in NOx concentrations in vehicle exhaust in our manuscript, which led to your misunderstanding. As a matter of fact, sampling of vehicle exhaust includes various kinds of sampling, i.e. direct sampling from the exhaust pipe, sampling at intersections where the road is congested, and sampling in tunnels where the traffic flow is high. Therefore, if it is expressed as a weighted average, it may instead introduce greater uncertainty.

L230-233: This considerable fluctuation is primarily due to the fact that the $NO_x$ samples from tunnels and intersections are the consequence of atmospheric dilution, which results in markedly lower concentrations than those obtained directly from vehicle exhausts (Text S1).

L244-246: However, the nitrogen content of gasoline is low, typically below 0.1% (Tang et al.,

2015), and the contribution of fuel NOx from vehicle exhaust is generally minimal in most cases.

L248-253: The differences in $\delta^{15}$N values of oils, influenced by their generation and depositional settings (Williams et al., 1995), lead to varying $\delta^{15}$N values of fuels NO$_x$. More importantly, the differing emission standards for vehicle exhaust across countries, result in local characteristics of $\delta^{15}$N values of NO$_x$ in vehicle exhaust (Zong et al., 2020a).

**Question 13:** Lines 228-229: Again, how is "significant" defined? Also, in the mentioned Figure S4b, it appears to have an error. Felix et al., 2012 report a $\delta^{15}$N(NO$_x$) from coal combustion up to 21.0 ‰ but this graph shows it only goes up to ~12‰. Please double-check this figure.

**Response 13:** Thanks for the comment. Inappropriate statements have been corrected in the revised version. In addition, the values in Figure S4b are means and standard deviations, not all ranges.

L256-257: Our results were differing from previous reported $\delta^{15}$N-NO$_x$ values from coal combustion (Figure S4b).

[Figure]

Fig S4 Comparison of $\delta^{15}$N values in NO$_x$ sources between this study and previous studies (Average ± standard deviation) (Fibiger and Hastings, 2016;Felix and Elliott, 2013, 2014;Felix et al., 2015;Felix et al., 2012;Li and Wang, 2008;Walters et al., 2015a;Walters et al., 2015b;Heaton, 1990;Miller et al., 2017;Zong et al., 2020a;Moore, 1977;Ammann et al., 1999;Redling et al.,

2013;Kawashima, 2019). Solid boxes indicate the mean value, and the left and right solid lines indicate the standard deviation.

*Question 14: Lines 247-249: In Walters et al., 2015a, both a residential furnace and a natural gas power plant were measured.*

**Response 14:** Thanks for the comment. Inappropriate statements have been corrected in the revised version.

L275-278: However, our results were more negative than those reported by Walters et al. (2015b) for $NO_x$ emitted from residential gas furnaces and natural gas power plant in Indiana, USA ( ranging from -19.7 ‰ to -13.9 ‰ and -16.5 ± 1.7 ‰).

*Question 15: Lines 323-326: Nitrate is noted to be positively correlated with $PM_{2.5}$. Please provide the correlation statistics.*

**Response 15:** Thanks for the comment. The correlation statistics were added in the revised version.

L354-357: The concentration of $NO_3^-$ showed a significant positive linear correlation with $PM_{2.5}$ (Figure 2) (r=0.8, $p < 0.01$), suggesting that the substantial increase in $PM_{2.5}$ pollution is linked to an increase in $NO_3^-$ concentration.

*Question 16: Lines 335-337: Was there a noticeable change in nitrate production mechanisms elucidated from $\delta^{18}O(NO_3^-)$ data?*

**Response 16:** Thanks for the comment. $\delta^{18}O\text{-}NO_3^-$ values similarly showed differences in $NO_3^-$ production mechanisms across sampling periods (Figure 2C and Figure S10). We have added specific analyses in the revised manuscript.

L410-414: As shown in Figure 2c, the $\delta^{18}O\text{-}NO_3^-$ values in this study ranged from 48.3‰ to 102.9‰, with a mean $\delta^{18}O$ value of 81.1 ± 11.5‰ (Table 2). Similar to $\delta^{15}N\text{-}NO_3^-$ values, the most positive $\delta^{18}O\text{-}NO_3^-$ value was observed during mid-heating (89.8 ± 9.9‰), followed by pre-heating (84.5 ± 8.4‰) and late-heating (73.0 ± 9.8‰) (Table 2).

L428-433: As shown in Figure S11, the contributions of •OH and $N_2O_5$ pathways were 35.4 ± 19.8% and 64.6 ± 19.8%,respectively, suggesting that $N_2O_5$ pathways dominates $NO_3^-$ formation. However, the contributions varied across different sampling periods, indicating that the pathway for $NO_3^-$ formation also varied. This finding aligns with the results presented in section 3.2.1.

*Question 17: Lines 366-369: The authors argue that the $\delta^{15}N(NO_3^-)$ differences between these three periods could be related to $NO_x$ source differences. However, what about isotope fractionation? They previously mentioned that the $NO_x$ oxidation efficiency changed as elucidated by $NO_3^-$ and $NO_2$ concentration trends (in which $NO_3^-$ concentrations didn't increase, but $NO_2$ did during the heating period). Including $\delta^{18}O(NO_3^-)$ data, as well as the fractionation corrected $\delta^{15}N(NO_x)$, would be important here to normalize the influence of potential chemical changes that might also impact $\delta^{15}N(NO_3^-)$.*

**Response 17:** Thanks for the comment. This less appropriate sentence has been corrected in the revised manuscript. In addition, we have added a discussion of isotope fractionation to the revised manuscript.

L400-401: These results mainly attributed to the variations in the sources of $NO_x$ during three sampling periods.

L410-438: As shown in Figure 2c, the $\delta^{18}O$-$NO_3^-$ values in this study ranged from 48.3‰ to 102.9‰, with a mean $\delta^{18}O$ value of 81.1 ± 11.5‰ (Table 2). Similar to $\delta^{15}N$-$NO_3^-$ values, the most positive $\delta^{18}O$-$NO_3^-$ value was observed during mid-heating (89.8 ± 9.9‰), followed by pre-heating (84.5 ± 8.4‰) and late-heating (73.0 ± 9.8‰) (Table 2). Furthermore, a significant positive linear correlation ($r = 0.7$, $p < 0.01$) was identified between $\delta^{15}N$-$NO_3^-$ and $\delta^{18}O$-$NO_3^-$, indicating that only two predominant oxidation pathways (•OH and $N_2O_5$ hydrolysis) govern $NO_3^-$ formation in this study (Xiao et al., 2020;Walters and Michalski, 2016). Previous studies have shown that the $\delta^{15}N$-$NO_3^-$ values in $PM_{2.5}$ does not fully reflect the initial $\delta^{15}N$-$NO_x$ due to the fractionation process between $NO_x$ and $NO_3^-$ (Fan et al., 2020;Song et al., 2019). Therefore, we calculated the initial $\delta^{15}N$-$NO_x$ values based on the $\delta^{15}N$ and $\delta^{18}O$ values of $NO_3^-$ as follows (Zong et al., 2017). First, the relative contributions of the •OH and $N_2O_5$ pathway were calculated separately using the $\delta^{18}O$-$NO_3^-$ values of $PM_{2.5}$. Second, the corresponding $\varepsilon N$ values ($\varepsilon_{•OH}$ and $\varepsilon_{N2O5}$) for the •OH and $N_2O_5$ pathways were estimated by considering the equilibrium isotopic fractionation between $NO_2$ and NO, and between $N_2O_5$ and $NO_2$, respectively. Finally, the $\varepsilon N$ value of $NO_x$ to $NO_3^-$ in $PM_{2.5}$ was calculated using the contributions of the two pathways and their corresponding $\varepsilon_{•OH}$ and $\varepsilon_{N2O5}$ values. The detailed procedures for all calculations can be found in the Supporting Information (Text 2). As shown in Figure S11, the contributions of •OH and $N_2O_5$ pathways were 35.4 ± 19.8% and 64.6 ± 19.8%, respectively, suggesting that $N_2O_5$ pathways dominates $NO_3^-$ formation. However, the contributions varied across different sampling periods, indicating that the pathway for $NO_3^-$ formation also varied. This finding aligns with the results presented in section 3.2.1. The calculated $\varepsilon N$ value of $NO_x$ to $NO_3^-$ were 7.5 ± 3.4‰ (Figure S12), close to the results of the previous studies in Beijing (Fan et al., 2020;Song et al., 2019), a large municipality near Tianjin. Furthermore, a slight difference in the $\varepsilon N$ value was found during different sampling periods (Figure S12), further indicating that isotopic fractionation similarly affects the feedback of $\delta^{15}N$-$NO_3^-$ to the $NO_x$ source.

*Question 18: Lines 415-419: Adding more parameters to an under-constrained mixing model will make the results more under-constrained. All the mixing model results have large standard deviations/uncertainties, and I'm not sure if it is warranted to discuss the differences between various model simulations, as all of the source estimates appear to overlap. For example, soil emissions are estimated to be as important as some combustion-related $NO_x$ emissions. This tends to invalidate the mixing model results, in my opinion, as these emissions should be rather low for a highly urbanized location during a cooler period.*

**Response 18:** Thanks for the comment. One advantage of using mixture models within a Bayesian framework is the ability to incorporate information from other data sources via informative prior distributions (Moore and Semmens, 2008). Once an informative prior for the proportional contributions of sources is established, MixSIAR can use this prior during model specification.

Prior beliefs about source contributions (fq) are defined using the Dirichlet distribution, with an interval of [0, 1], and the total contributions of all sources default to 1. For instance, if only three sources are input into the model, it will assess their contributions assuming their total equals 1. If the sum of these contributions is significantly less than 1 in a mixed environment, additional sources must be included to accurately estimate each source's contribution. Thus, omitting necessary source input models can increase uncertainty in the model's quantitative results. We recognize the reviewers' concerns about the uncertainty associated with multiple source contributions. In fact, Fan et al. (2020) demonstrated that adding irrelevant sources does not significantly affect the regularity of source resolution results. Specifically, during the cold winter months in Beijing, soil source inputs had little effect on the quantitative results of the model. In this study, the wide variation in the results resolved by the models in our multiple scenarios rather highlights the need for our study. Moreover, the posterior distributions and stability for the proportional contribution of each Scenarios were compared (Figure 4 and Table S2). In result, as the number of sources input to the model increased, the contribution of various $NO_x$ sources was becoming more stable, and the inter-influence between various sources significantly reduced. This implied that is no significant interinfluence in terms of estimated source apportionments when the more emission sources were considered in SIAR model.

Additionally, the reviewers expressed skepticism regarding the contribution of soil sources in our results. While Tianjin is an urban environment, it is situated in the North China Plain and is surrounded by extensive agricultural land. Furthermore, the urban area includes a significant amount of green belt soil, which remains vegetated even in winter. It is important to note that temperatures were close to 10°C during both sampling periods, except for the mid-heating period, suggesting that microbial activity was still active. Under these conditions, more NOx from biogenic soil emissions would be expected to enter the atmosphere due to the relatively high soil temperature (ambient temperature: ~10°C) (Williams et al., 1992). Thus, it is not surprising that the mean contribution of soil sources was 10% during the sampling periods (Figure 5).

*Question 19:* *Lines 585-589: Again, I think weighted averages might be better to report here than unweighted.*

**Response 19:** Thanks for the comment. We have already responded to this question in *Question 12*. That is, sampling of vehicle exhaust includes various kinds of sampling, i.e. direct sampling from the exhaust pipe, sampling at intersections where the road is congested, and sampling in tunnels where the traffic flow is high. Therefore, if it is expressed as a weighted average, it may instead introduce greater uncertainty.

*Question 20:* *Table 1. Walters et al., 2015b did not report coal combustion $\delta^{15}N(NO_x)$. For soil emissions, you may also consider adding (Yu and Elliott, 2017).*

**Response 20:** Thanks for the comment. This error has been corrected.

Table 1 Comparison of the $\delta^{15}N$ characteristic spectra of $NO_x$ sources reported in the previous and present study

| $NO_x$ sources | Previous studies | This study |
|---|---|---|

|  | av (‰) | std (‰) | n | Reference | av (‰) | std (‰) | n |
|---|---|---|---|---|---|---|---|
| Coal combustion | +13.7 | 4.6 | 47 | (Felix et al., 2012;Heaton, 1990) | +12.3 | 1.7 | 13 |
| Vehicle exhaust | −7.2 | 7.8 | 151 | (Walters et al., 2015b;Walters et al., 2015a;Felix and Elliott, 2014;Heaton, 1990) | −5.2 | 5.4 | 62 |
| Biomass burning | +1.0 | 4.1 | 24 | (Fibiger and Hastings, 2016;Felix and Elliott, 2013) | +1.2 | 3.0 | 14 |
| Soil emission | −33.8 | 12.2 | 6 | (Felix and Elliott, 2014;Li and Wang, 2008;Yu and Elliott, 2017) | −33.7 | 9.7 | 12 |
| Nature gas combustion | −16.5 | 1.7 | 23 | (Walters et al., 2015b) | −24.8 | 5.6 | 5 |
| Industrial source | N/A | N/A | | N/A | −20.6 | 16.8 | 17 |

Note:N/A represents data unknown.

***Question 21:*** *Fig 3. It would be helpful if the stacked bar plots followed the legend order for easier visual interpretation. Furthermore, it appears that this is an error in the figure caption, as Scenarios 1 and 3 are defined twice, while Scenarios 2 and 4 are undefined.*

**Response 21:** Thanks for the comment. This error has been corrected.

[Figure]

Figure 3 Comparison of fractional contributions of $NO_3^-$ sources in $PM_{2.5}$ in Tianjin estimated by different $\delta^{15}N$ values of $NO_x$ sources. The results of Scenario 1 and Scenario 3 were estimated

using the $\delta^{15}$N values of four and five NO$_x$ sources obtained from previous studies, while the results of Scenario 2 and Scenario 4 were estimated using the $\delta^{15}$N values of four and five NO$_x$ sources obtained from this study. Also, SE = soil emission, CC = coal combustion, BB = biomass burning, VE = vehicle emission, and CG = combustion of natural gas.

**Technical Corrections:**

Technical Comment 1: Through the manuscript, please change NOx to NOx (with the "x" formatted as a subscript and in italics).

**Response 1:** Thanks for the comment. Similar problems have all been corrected in the revised manuscript.

Technical Comment 2: Please write all quantity symbols (including δ, n, etc) in italics.

**Response 2:** Thanks for the comment. Similar problems have all been corrected in the revised manuscript.

Technical Comment 3: Lines 59-61, this is an incomplete sentence.

**Response 3:** Thanks for the comment. We have rewritten the sentence.

L59-61: It should be noted that the $\delta^{15}$N values from various NO$_x$ sources have been reported in previous studies (Zong et al., 2020a;Zong et al., 2022a).

Technical Comment 4: Lines 138-141: You can delete the word "Initially" here.

**Response 4:** Thanks for the comment. The word "Initially" has been deleted.

L149-152: Hydrophobic Teflon membrane (TF-200, Pall, USA) and nylon membrane (BNRG810S, Pall, USA) were······

Technical Comment 5: Lines 156-157: Please change "cutted" to "cut"

**Response 5:** Thanks for the comment. This error has been corrected.

L167-168: ······a proportion of the particulate matter from the each PM$_{2.5}$ sample was cut and······

Technical Comment 6: Lines 166-170: Neutralized is used twice in this sentence. I would suggest deleting the "to neutralize…" part of the sentence.

**Response 6:** Thanks for the comment. This error has been corrected.

L177-180: ······which was neutralized using electronic grade hydrochloric acid (HCl) at a mass concentration of approximately 38%.

Technical Comment 7: Lines 170-171: You can delete "salt" from this sentence

**Response 7:** Thanks for the comment. "salt" has been deleted in this sentence.

L180-181: Finally, the supernatant was analyzed for $NO_2^-$ and $NO_3^-$ concentrations on a Skalar San++ continuous flow nutrient analyzer.

Technical Comment 8: Lines 179: "Pseudomonas aureofaciens" should be italicized.

**Response 8:** Thanks for the comment. We have corrected it.

L188-190: Briefly, extracted $NO_3^-$ were quantitatively converted to $N_2O$ through the action of denitrifying bacteria, namely *Pseudomonas aureofaciens*, ATCC 13985, which lacked $N_2O$ reductase (Luo et al., 2020).

Technical Comment 9: Text S2: Nitrate-forming reactions aren't typically expected to be "aqueous" reactions, such as in cloud reactions. Instead, I think the authors mean heterogeneous, and I would recommend switching these terms.

**Response 9:** Thanks for the comment. We have corrected it.

Text S2: The formation pathways of emitted $NO_x$ (NO+NO₂) to $NO_3^-$ in polluted cities are complex, which included heterogeneous and gas-phase reactions.

Technical Comment 10: Line 222: "Significantly" is used here and in other places in the manuscript. What significant test was utilized and what are the p-values to indicate significance?

**Response 10:** Thanks for the comment. Significance test methods and p-values have been added in the revised manuscript.

L246-248: ……our results align with domestic reports but significantly differ from foreign studies (Fisher's least significant difference (LSD) test, the same as below; $p < 0.05$).

Technical Comment 11: Lines 363-366: Please provide p-values to indicate whether these differences are significant.

**Response 11:** Thanks for the comment. The p-values have been added in the revised manuscript.
L397-398: Significant differences were observed in $\delta^{15}N\text{-}NO_3^-$ values among the three sampling periods in this study ($p < 0.05$).

**References**

Bekker, C., Walters, W. W., Murray, L. T., and Hastings, M. G.: Nitrate chemistry in the northeast US–Part 1: Nitrogen isotope seasonality tracks nitrate formation chemistry, Atmospheric Chemistry and Physics, 23, 4185–4201, 2023.

Ammann, M., Siegwolf, R., Pichlmayer, F., Suter, M., Saurer, M., and Brunold, C.: Estimating the

uptake of traffic-derived $NO_2$ from [15]N abundance in Norway spruce needles, Oecologia, 118, 124-131, 10.1007/s004420050710, 1999.

Böhlke, J. K., Mroczkowski, S. J., and Coplen, T. B.: Oxygen isotopes in nitrate: new reference materials for $^{18}O:^{17}O:^{16}O$ measurements and observations on nitrate-water equilibration, Rapid. Commun. Mass. Sp., 17, 1835-1846, https://doi.org/10.1002/rcm.1123, 2003.

Casciotti, K. L., Sigman, D. M., Hastings, M. G., Böhlke, J. K., and Hilkert, A.: Measurement of the oxygen isotopic composition of nitrate in seawater and freshwater using the denitrifier method, Anal. Chem., 74, 4905-4912, https://doi.org/10.1021/ac020113w, 2002.

Fan, M., Zhang, Y., Lin, Y., Cao, F., Zhao, Z., Sun, Y., Qiu, Y., Fu, P., and Wang, Y.: Changes of emission sources to nitrate aerosols in Beijing after the clean air actions: evidence from dual isotope compositions, J. Geophys. Res. Atmos., 125, 1-15, https://doi.org/10.1029/2019JD031998, 2020.

Fang, H., Walters, W. W., Mase, D., and Michalski, G.: iNRACM: incorporating [15]N into the regional atmospheric chemistry mechanism (RACM) for assessing the role photochemistry plays in controlling the isotopic composition of NOx, NOy, and atmospheric nitrate, Geosci. Model Dev., 14, 5001-5022, 10.5194/gmd-14-5001-2021, 2021.

Felix, J. D., Elliott, E. M., and Shaw, S. L.: Nitrogen isotopic composition of coal-fired power plant $NO_x$: influence of emission controls and implications for global emission inventories, Environ. Sci. Technol., 46, 3528-3535, https://doi.org/10.1021/es203355v, 2012.

Felix, J. D., and Elliott, E. M.: The agricultural history of human-nitrogen interactions as recorded in ice core $\delta^{15}N$-$NO_3^-$, Geophys. Res. Lett., 40, 1642-1646, https://doi.org/10.1002/grl.50209, 2013.

Felix, J. D., and Elliott, E. M.: Isotopic composition of passively collected nitrogen dioxide emissions: Vehicle, soil and livestock source signatures, Atmos. Environ., 92, 359-366, https://doi.org/10.1016/j.atmosenv.2014.04.005, 2014.

Felix, J. D., Elliott, E. M., Avery, G. B., Kieber, R. J., Mead, R. N., Willey, J. D., and Mullaugh, K. M.: Isotopic composition of nitrate in sequential Hurricane Irene precipitation samples: Implications for changing NOx sources, Atmos. Environ., 106, 191-195, https://doi.org/10.1016/j.atmosenv.2015.01.075, 2015.

Feng, X., Chen, Y., Du, H., Feng, Y., Mu, Y., and Chen, J.: Biomass burning is a non-negligible source for ammonium during winter haze episodes in rural North China: Evidence from high time resolution [15]N-stable isotope, J. Geophys. Res. Atmos., 128, e2022JD038012, https://doi.org/10.1029/2022JD038012, 2023.

Fibiger, D. L., and Hastings, M. G.: First measurements of the nitrogen isotopic composition of NOx from biomass burning, Environ. Sci. Technol. , 50, 11569-11574, https://doi.org/10.1021/acs.est.6b03510, 2016.

Heaton, T. H. E.: $^{15}N/^{14}N$ ratios of $NO_x$ from vehicle engines and coal-fired power stations, Tellus, 42, 304-307, https://doi.org/10.1034/j.1600-0889.1990.00007.x-i1, 1990.

Jin, Z., Li, J., Yang, Q., Shi, Y., Lin, X., Chen, F., Chen, Q., Chen, Z., and Li, F.: Nitrogen isotope characteristics and importance of NOx from biomass burning in China, Sci. Tot. Environ., 951, 175430, https://doi.org/10.1016/j.scitotenv.2024.175430, 2024.

Kawashima, H.: Seasonal trends of the stable nitrogen isotope ratio in particulate nitrogen compounds and their gaseous precursors in Akita, Japan, Tellus B, 71, 1-13, https://doi.org/10.1080/16000889.2019.1627846, 2019.

Li, D., and Wang, X.: Nitrogen isotopic signature of soil-released nitric oxide (NO) after fertilizer

application, Atmos. Environ., 42, 4747-4754, https://doi.org/10.1016/j.atmosenv.2008.01.042, 2008.

Li, J., Zhang, X., Orlando, J., Tyndall, G., and Michalski, G.: Quantifying the nitrogen isotope effects during photochemical equilibrium between NO and $NO_2$: implications for $\delta^{15}N$ in tropospheric reactive nitrogen, Atmos. Chem. Phys., 20, 9805-9819, 10.5194/acp-20-9805-2020, 2020.

Li, Q., Li, X., Yang, Z., Cui, G., and Ding, S.: Diurnal and seasonal variations in water-soluble inorganic ions and nitrate dual isotopes of $PM_{2.5}$: Implications for source apportionment and formation processes of urban aerosol nitrate, Atmos. Res., 248, 105197, https://doi.org/10.1016/j.atmosres.2020.105197, 2021.

Li, T., Li, J., Sun, Z., Jiang, H., Tian, C., and Zhang, G.: High contribution of anthropogenic combustion sources to atmospheric inorganic reactive nitrogen in South China evidenced by isotopes, Atmos. Chem. Phys., 23, 6395-6407, https://doi.org/10.5194/acp-23-6395-2023, 2023.

Li, Y., Geng, Y., Hu, X., and Yin, X.: Seasonal differences in sources and formation processes of $PM_{2.5}$ nitrate in an urban environment of North China, J. Environ. Sci., 120, 94-104, https://doi.org/10.1016/j.jes.2021.08.020, 2022.

Lin, Y.-C., Zhang, Y.-L., Yu, M., Fan, M.-Y., Xie, F., Zhang, W.-Q., Wu, G., Cong, Z., and Michalski, G.: Formation mechanisms and source apportionments of airborne nitrate aerosols at a Himalayan-Tibetan Plateau site: Insights from nitrogen and oxygen isotopic compositions, Environ. Sci. Technol., 55, 12261-12271, https://doi.org/10.1021/acs.est.1c03957, 2021.

Lin, Y.-C., Fan, M.-Y., Hong, Y., Yu, M., Cao, F., and Zhang, Y.-L.: Important contributions of natural gas combustion to atmospheric nitrate aerosols in China: Insights from stable nitrogen isotopes, Sci. Bull., https://doi.org/10.1016/j.scib.2024.06.038, 2024.

Luo, L., Wu, Y., Xiao, H., Zhang, R., Lin, H., Zhang, X., and Kao, S.: Origins of aerosol nitrate in Beijing during late winter through spring, Sci. Tot. Environ., 653, 776-782, https://doi.org/10.1016/j.scitotenv.2018.10.306, 2019.

Luo, L., Zhu, R. G., Song, C. B., Peng, J. F., and Xiao, H. Y.: Changes in nitrate accumulation mechanisms as $PM_{2.5}$ levels increase on the North China Plain: A perspective from the dual isotopic compositions of nitrate, Chemosphere, 263, 127915, https://doi.org/10.1016/j.chemosphere.2020.127915, 2020.

Miller, D. J., Wojtal, P. K., Clark, S. C., and Hastings, M. G.: Vehicle NO emission plume isotopic signatures: Spatial variability across the eastern United States, J. Geophys. Res. Atmos., 122, 4698-4717, https://doi.org/10.1002/2016JD025877, 2017.

Moore, H.: The isotopic composition of ammonia, nitrogen dioxide and nitrate in the atmosphere, Atmos. Environ., 11, 1239-1243, https://doi.org/10.1016/0004-6981(77)90102-0, 1977.

Moore, J. W., and Semmens, B. X.: Incorporating uncertainty and prior information into stable isotope mixing models, Ecol. Lett., 11, 470-480, https://doi.org/10.1111/j.1461-0248.2008.01163.x, 2008.

Parnell, A. C., Inger, R., Bearhop, S., and Jackson, A. L.: Source partitioning using stable isotopes: Coping with too much variation, PLOS ONE, 5, e9672, https://doi.org/10.1371/journal.pone.0009672, 2010.

Parnell, A. C., Phillips, D. L., Bearhop, S., Semmens, B. X., Ward, E. J., Moore, J. W., Jackson, A. L., Grey, J., Kelly, D. J., and Inger, R.: Bayesian stable isotope mixing models, Environmetrics, 24, 387-399, https://doi.org/10.1002/env.2221, 2013.

Redling, K., Elliott, E., Bain, D., and Sherwell, J.: Highway contributions to reactive nitrogen deposition: tracing the fate of vehicular NOx using stable isotopes and plant biomonitors, Biogeochemistry, 116, 261-274, 10.1007/s10533-013-9857-x, 2013.

Shang, X., Huang, H., Mei, K., Xia, F., Chen, Z., Yang, Y., Dahlgren, R. A., Zhang, M., and Ji, X.: Riverine nitrate source apportionment using dual stable isotopes in a drinking water source watershed of southeast China, Sci. Tot. Environ., 724, 137975, https://doi.org/10.1016/j.scitotenv.2020.137975, 2020.

Sigman, D. M., Casciotti, K. L., Andreani, M., Barford, C., Galanter, M., and Böhlke, J. K.: A bacterial method for the nitrogen isotopic analysis of nitrate in seawater and freshwater, Anal. Chem., 73, 4145-4153, https://doi.org/10.1021/ac010088e, 2001.

Song, W., Wang, Y., Yang, W., Sun, X., Tong, Y., Wang, X., Liu, C., Bai, Z., and Liu, X.: Isotopic evaluation on relative contributions of major NOx sources to nitrate of $PM_{2.5}$ in Beijing, Environ. Pollut., 248, 183-190, https://doi.org/10.1016/j.envpol.2019.01.081, 2019.

Stock, B. C., Jackson, A. L., Ward, E. J., Parnell, A. C., Phillips, D. L., and Semmens, B. X.: Analyzing mixing systems using a new generation of Bayesian tracer mixing models, PeerJ, 6, 2167-8359 2018.

Tang, G., Sun, J., Wu, F., Sun, Y., Zhu, X., Geng, Y., and Wang, Y.: Organic composition of gasoline and its potential effects on air pollution in North China, Sci. China Chem., 58, 1416-1425, https://doi.org/10.1007/s11426-015-5464-0, 2015.

Walters, W. W., Goodwin, S. R., and Michalski, G.: Nitrogen stable isotope composition ($\delta^{15}$N) of vehicle-emitted NOx, Environ. Sci. Technol., 49, 2278-2285, https://doi.org/10.1021/es505580v, 2015a.

Walters, W. W., and Michalski, G.: Theoretical calculation of nitrogen isotope equilibrium exchange fractionation factors for various $NO_y$ molecules, Geochim. Cosmochim. Acta., 164, 284-297, https://doi.org/10.1016/j.gca.2015.05.029, 2015.

Walters, W. W., Tharp, B. D., Fang, H., Kozak, B. J., and Michalski, G.: Nitrogen isotope composition of thermally produced NOx from various fossil-fuel combustion sources, Environ. Sci. Technol., 49, 11363-11371, https://doi.org/10.1021/acs.est.5b02769, 2015b.

Walters, W. W., and Michalski, G.: Theoretical calculation of oxygen equilibrium isotope fractionation factors involving various $NO_y$ molecules, radical OH, and $H_2O$ and its implications for isotope variations in atmospheric nitrate, Geochim. Cosmochim. Acta., 191, 89-101, https://doi.org/10.1016/j.gca.2016.06.039, 2016.

Walters, W. W., Karod, M., Willcocks, E., Baek, B. H., Blum, D. E., and Hastings, M. G.: Quantifying the importance of vehicle ammonia emissions in an urban area of the northeastern US utilizing nitrogen isotopes, Atmos. Chem. Phys. , 22, 13431–13448, https://doi.org/10.5194/acp-2022-419, 2022.

Williams, E. J., Hutchinson, G. L., and Fehsenfeld, F. C.: NOx and $N_2O$ emissions from soil, Global Biogeochem Cy., 6, 351-388, https://doi.org/10.1029/92GB02124, 1992.

Williams, L. B., Ferrell, R. E., Hutcheon, I., Bakel, A. J., Walsh, M. M., and Krouse, H. R.: Nitrogen isotope geochemistry of organic matter and minerals during diagenesis and hydrocarbon migration, Geochim. Cosmochim. Ac., 59, 765-779, https://doi.org/10.1016/0016-7037(95)00005-K, 1995.

Xiao, H., Ding, S. Y., Ji, C. W., Li, Q. K., and Li, X. D.: Strict control of biomass burning inhibited particulate matter nitrate pollution over Tianjin: Perspective from dual isotopes of nitrate, Atmos. Environ., 293, 119460, https://doi.org/10.1016/j.atmosenv.2022.119460, 2023.

Xiao, H. W., Zhu, R. G., Pan, Y. Y., Guo, W., Zheng, N. J., Liu, Y. H., Liu, C., Zhang, Z. Y., Wu, J. F., Kang, C. A., Luo, L., and Xiao, H. Y.: Differentiation between nitrate aerosol formation pathways in a Southeast Chinese city by dual isotope and modeling studies, J. Geophys. Res. Atmos., 125, e2020JD032604, https://doi.org/10.1029/2020jd032604, 2020.

Yu, Z., and Elliott, E. M.: Novel method for nitrogen isotopic analysis of soil-emitted nitric oxide, Environ. Sci. Technol., 51, 6268-6278, https://doi.org/10.1021/acs.est.7b00592, 2017.

Zhang, W., Wu, F., Luo, X., Song, L., Wang, X., Zhang, Y., Wu, J., Xiao, Z., Cao, F., Bi, X., and Feng, Y.: Quantification of NOx sources contribution to ambient nitrate aerosol, uncertainty analysis and sensitivity analysis in a megacity, Sci. Tot. Environ., 926, 171583, https://doi.org/10.1016/j.scitotenv.2024.171583, 2024.

Zhang, Z., Zheng, N., Zhang, D., Xiao, H., and Xiao, H.: Rayleigh based concept to track $NO_x$ emission sources in urban areas of China, Sci. Tot. Environ., 704, 135362, https://doi.org/10.1016/j.scitotenv.2019.135362, 2019.

Zhao, Z., Cao, F., Fan, M., Zhang, W., Zhai, X., Wang, Q., and Zhang, Y.: Coal and biomass burning as major emissions of $NO_x$ in Northeast China: Implication from dual isotopes analysis of fine nitrate aerosols, Atmos. Environ., 242, 117762, https://doi.org/10.1016/j.atmosenv.2020.117762, 2020.

Zhao, Z. Y., Cao, F., Fan, M. Y., Zhai, X. Y., Yu, H. R., Hong, Y., Ma, Y. J., and Zhang, Y. L.: Nitrate aerosol formation and source assessment in winter at different regions in Northeast China, Atmos. Environ., 267, 118767, https://doi.org/10.1016/j.atmosenv.2021.118767, 2021.

Zong, Z., Wang, X., Tian, C., Chen, Y., Fang, Y., Zhang, F., Li, C., Sun, J., Li, J., and Zhang, G.: First assessment of $NO_x$ sources at a regional background site in North China using isotopic analysis linked with modeling, Environ. Sci. Technol., 51, 5923-5931, https://doi.org/10.1021/acs.est.6b06316, 2017.

Zong, Z., Sun, Z., Xiao, L., Tian, C., Liu, J., Sha, Q., Li, J., Fang, Y., Zheng, J., and Zhang, G.: Insight into the variability of the nitrogen isotope composition of vehicular NOx in China, Environ. Sci. Technol., 54, 14246-14253, https://doi.org/10.1021/acs.est.0c04749, 2020a.

Zong, Z., Tan, Y., Wang, X., Tian, C., Li, J., Fang, Y., Chen, Y., Cui, S., and Zhang, G.: Dual-modelling-based source apportionment of $NO_x$ in five Chinese megacities: Providing the isotopic footprint from 2013 to 2014, Environ. Int., 137, 105592, https://doi.org/10.1016/j.envint.2020.105592, 2020b.

Zong, Z., Shi, X., Sun, Z., Tian, C., Li, J., Fang, Y., Gao, H., and Zhang, G.: Nitrogen isotopic composition of NOx from residential biomass burning and coal combustion in North China, Environ. Pollut., 304, 119238, https://doi.org/10.1016/j.envpol.2022.119238, 2022a.

Zong, Z., Tian, C., Sun, Z., Tan, Y., Shi, Y., Liu, X., Li, J., Fang, Y., Chen, Y., Ma, Y., Gao, H., Zhang, G., and Wang, T.: Long-term evolution of particulate nitrate pollution in North China: Isotopic evidence from 10 offshore cruises in the Bohai Sea from 2014 to 2019, J. Geophys. Res. Atmos., e2022JD036567, https://doi.org/10.1029/2022JD036567, 2022b.

---

## Author Comment (AC2)

**Detailed response to the reviewers' comments**

Atmospheric Chemistry and Physics (Paper egusphere-2024-1621)

Title: "Technical note: Refining $\delta^{15}$N isotopic fingerprints of local NO$_x$ for accurate source identification of nitrate in PM$_{2.5}$"

Dear Editor(s) and reviewers,

We would like to express our gratitude for your valuable comments and suggestions. The manuscript has been carefully revised according to the comments.

Response to the editor(s)' comments and detailed point-by-point responses to the reviewers' comments are provided in the following pages. Please note that the comments are presented in *italics font*, and our responses are in Roman font and marked in blue. In addition, all the line numbers in the responses refer to the revised manuscript. All changes made to the manuscript are marked in red.

Please let us know if you have any further questions regarding our manuscript.

We are looking forward to your reply.

Sincerely,

Pro. Xiaodong Li

E-mail: xiaodong.li@tju.edu.cn

Institute of Surface-Earth System Science, School of Earth System Science

Tianjin University, Tianjin, P. R. China, 300072

**Reviewer #2:**

*This work reports the measurement of the nitrogen isotopes ($\delta^{15}N$) from major domestic $NO_x$ sources such as vehicle exhaust, industrial emissions, and natural gas combustion in Tianjin, China. The authors directly collected $\delta^{15}N$ samples from many NOx sources using active sampling methods and compared their values with previous studies. Different $\delta^{15}N(NO_x)$ values from the previous studies were obtained in this area due to the local characteristics of $NO_x$ sources. They also measured inorganic ions of $PM_{2.5}$ over three different heating periods to understand the contribution of $NO_x$ to nitrate formation in Tianjin. Coal combustion is reported as the main source of $NO_3^-$ during heating periods in this area. In addition, the contributions from different $NO_x$ sources to $NO_3^-$ in $PM_{2.5}$ were quantified using a stable isotope mixing model depending on many different scenarios. In general, the study primarily focuses on refining the $\delta^{15}N(NO_x)$ values from various $NO_x$ sources in Tianjin, China, suggesting that these values reflect the local characteristics. While I appreciate the effort to collect and determine $\delta^{15}N(NO_x)$ values from various $NO_x$ sources and quantify $NO_3^-$ formation in China, I believe several issues should be addressed and further discussion should be added when considering this study for publication in the ACP.*

*First, the manuscript type of technical notes in the ACP is meant to be peer-reviewed publications that report new developments, significant advances, or novel aspects of experimental and theoretical methods and techniques -- I feel like this study doesn't meet the standard of ACP publication as a technical note. Rather, it looks closer to a measurement report since they mainly reported the measured $\delta^{15}N(NO_x)$ values depending on the $NO_x$ sources and inorganic ions of $PM_{2.5}$. Additionally, their major discussion focuses on the isotope mixing model to estimate the contribution of $NO_x$ sources, but their outputs are not validated. The isotope mixing model is commonly used in the isotope field to identify and quantify the contributions of multiple sources using potential source end-members. However, this statistical approach has limitations in accurately quantifying source portions. In this study, the measurement values used as inputs show high standard deviations and wide ranges. Given the isotope mixing model is sensitive to input parameters, large variabilities can lead to significant uncertainties in the output. Further, there are known large fractionations of nitrogen isotope between $NO_x$ to $NO_3^-$ conversion (Li et al., 2020; Bekker et al., 2023), but these isotope effects are not considered in this study. It is also unclear which values are used for the mixing model. The model output should be validated to ensure its reliability through NOx emission inventory data or other types of observations from the Tianjin area. Lastly, the authors mentioned that a systematic study of $\delta^{15}N(NO_x)$ values from domestic $NO_x$ sources is crucial for accurately identifying nitrate sources (Lines 15-16). The distinct values obtained in this study reflect the local characteristics of $NO_x$ emissions, indicating that these $\delta^{15}N(NO_x)$ values might have limitations when applied to other regions. Therefore, it would be helpful if the author provided recommendations on how to best use these values and clarified their scientific relevance to other regions.*

**Response**: We thank the reviewers for investing the time to thoroughly evaluate our initial manuscript and for the constructive comments.

    **First,** the reviewers suggested that our study resembles a measurement report. Initially, our

manuscript was indeed submitted to "Measurement Report." However, since the focus is on an enhanced framework for identifying aerosol nitrate sources through stable nitrogen isotopic analysis of local sources, the editor recommended that the paper be considered a "Technical Note." We are open to publishing the manuscript as a "Measurement Report" if feasible.

**Second**, reviewers were concerned about the results of our model quantification. One of the benefits to conducting mixture models in a Bayesian framework is that information from other data sources can be included via informative prior distributions (Parnell et al., 2013;Moore and Semmens, 2008). Once an informative prior for the proportional contribution of sources is established, MixSIAR can accept the prior as an input during the model specification process. Although some of them propose to the results may be inaccurate when utilizing one variable ($\delta^{15}$N) to mix between several parameters (4 or 5 sources). Generally, prior beliefs about proportional source contributions ($f_q$) are defined using the Dirichlet distribution, with an interval of [0, 1], and the sum of the contributions of the sources of all input models defaults to 1. For example, if only 3 sources are input to the model, the model will evaluate the source contributions based on the sum of their contributions being 1. However, if the sum of the contributions from these three sources is much less than 1 in a mixed environment, then data from more sources will need to be entered to more accurately estimate the corresponding contribution from each source. In other words, the absence of the number of source input models may lead to increased uncertainty in the quantitative results of the model. More important, a growing number of recent studies have suggested the need to increase the number of sources in the model to eliminate the interactions between the various sources (Lin et al., 2024;Lin et al., 2021;Zhang et al., 2024). In the SIAR model, the Monte Carlo approach was used to quantify the emission sources of nitrate aerosols (Zong et al., 2017), it is also widely used in similar studies (Wu et al., 2024;Cheng et al., 2022;Chen et al., 2022;Song et al., 2021). To enhance the reliability of the study results, the model generated 10,000 potential scenarios for each evaluated potential source (Song et al., 2019;Fan et al., 2020). Finally, the posterior distributions and stability for the proportional contribution of each Scenarios were compared (Figure 4 and Table S2). In result, as the number of sources input to the model increased, the contribution of various NOx sources was becoming more stable, and the inter-influence between various sources significantly reduced. This implied that is no significant interinfluence in terms of estimated source apportionments when the more emission sources were considered in SIAR model. Overall, we believe that the results quantified by the model in this study are acceptable.

**Third**, the effect of the fractionations of nitrogen isotope between $NO_x$ to $NO_3^-$ conversion has been added in the revised version. In this study, the $\delta^{18}$O-$NO_3^-$ values helped constrain the fractionation factor from $NO_x$ to $NO_3^-$ (Xiao et al., 2020), but only two primary pathways, hydroxyl radical oxidation and nitrogen pentoxide hydrolysis, were taken into account. Previous research supports the view that these pathways account for up to 95% of $NO_3^-$ production (Lin et al., 2021;Xiao et al., 2020), implying that alternative pathways might exert a relatively minor impact on $\varepsilon$N calculations. Nonetheless, future measurements of $\Delta^{17}$O- $NO_3^-$ are essential to elucidate the isotopic fractionation coefficients comprehensively during the formation of $NO_3^-$.

**Finally,** we acknowledge the limitations of our study and offer recommendations for future research. We analyzed only six samples from typical $NO_x$ sources in Tianjin, without examining all $NO_x$ sources in China. Variations in $\delta^{15}$N background values, combustion processes, and $NO_x$ emission standards affect the $\delta^{15}$N signal of $NO_x$ emissions from different sources. Given the

uniformity of industrial standards across Chinese cities, these values could also help determine the source of $NO_3^-$ in other locations. Our study emphasizes the need for a comprehensive determination of $\delta^{15}N$ values for typical $NO_x$ sources. Refining $NO_x$ source types and improving $\delta^{15}N$ values for other $NO_x$ sources would enhance $NO_3^-$ source apportionment in future research. Besides, the absence of constraints in the model may introduce some uncertainty into the results of this study. Consequently, further refinement may be necessary in the future to address this issue. Relevant content can be found in section 3.5 of the revised manuscript.

We have carefully modified and proofread the manuscript. Below, we will provide detailed one-on-one revisions and responses to deficiencies in the original manuscript. All changes are marked by the red font in the revised manuscript.

**Major Comments:**

*Question 1: This manuscript aims to explore $NO_x$ emission sources in the Tianjin area using isotope analysis. The author mentioned in the introduction that this area predominantly has high $NO_x$ emissions from coal combustion for heating during the heating period (Lines 93-94). However, there is no mention of other periods, such as preheating and late heating, or other potential emission sources, even though the highest $NO_3^-$ and $PM_{2.5}$ concentrations were observed during the preheating period. What would be the potential sources of $NO_3^-$ and $PM_{2.5}$ during these three different periods? It would be helpful if this argument could be supported or compared with $NO_x$ emission inventory data or another type of observation from the Tianjin area.*

**Response 1:** Thanks for the comment. Apologies for our unclear presentation. We have cited previous studies to confirm the significant contribution of coal combustion during heating in Tianjin. However, it is important to note that coal combustion only contributes a higher proportion during the heating period, rather than a change in the overall source structure. In other words, only the proportion of the source contribution changed during the three sampling periods, not the source structure.

L97-99: For instance, previous studies have demonstrated that $NO_2$ emissions in the Beijing-Tianjin-Hebei region exhibit a marked increase during the heating season, reaching an annual peak in winter (Meng et al., 2018).

*Question 2: In this study, $PM_{2.5}$ samples were collected for the day and night time from Oct to Apr, but the diurnal or seasonal pattern of $NO_x$ and $NO_3^-$ are not considered at all in this study. Meteorological factors, especially temperature, have a significant effect on nitrogen isotope fractionation and nitrate formation. Further, the major sources of $NO_x$ emission could be variable depending on meteorological conditions. How do the meteorological factors affect the $NO_x$ and $NO_3^-$ in this study?*

**Response 2:** Thanks for the comment. The change pattern of $NO_x$ and $NO_3^-$ are considered in this study. In addition, we have added a discussion of the effects of meteorological parameters on $NO_3^-$ concentrations. In conclusion, the combination of $NO_2$ emissions and variations in meteorological parameters, including wind speed and relative humidity, contributed to the observed differences in nitrate concentrations between sampling periods in this study. It is

important to note that the main purpose of this study is to discuss the importance in refined isotopic fingerprint of $NO_x$ source in a region-specific context. Consequently, the seasonal pattern of $NO_x$ and $NO_3^-$, along with the influence of meteorological factors, was only superficially examined.

L358-384: The variation pattern of $NO_3^-$ concentration during different sampling periods aligned with that of $PM_{2.5}$ (Figure 2). The highest concentration of $NO_3^-$ was observed during pre-heating ($16.0 \pm 12.4$ µg m$^{-3}$), and the lowest concentration was observed during late-heating ($9.7 \pm 8.7$ µg m$^{-3}$) (Table 2). This pattern of change is associated with the discrepancy in wind speed (Table S1), as higher wind speeds may not be conducive to $NO_3^-$ accumulation. That is, the mean wind speed is lowest during pre-heating periods, resulting in greater $NO_3^-$ accumulation in the atmosphere. Notably, $NO_2$, as the precursor of $NO_3^-$, did not follow the observed pattern of change in $NO_3^-$ concentration. The highest concentration of $NO_2$ was observed during mid-heating (Table S1 and Figure S7), potentially influenced by increased coal combustion for heating (Luo et al., 2019). Generally, $NO_2$ concentration and its secondary conversion efficiency were the key factors affecting the concentration of $NO_3^-$. Therefore, the difference could be attributed to biases in the secondary conversion efficiency of $NO_2$ (Xiao et al., 2023). This is also corroborated by the considerable discrepancy between the sampling periods in the key meteorological factors influencing nitrate formation, such as relative humidity. Compared to $NO_3^-$, the $SO_4^{2-}$ concentration was highest during mid-heating. In addition, $SO_2$, mainly originating from coal combustion, exhibited a similar variation pattern to $SO_4^{2-}$, potentially attributed to increased coal combustion for heating (Figure S9) (Feng et al., 2020). Markers primarily originating from coal combustion, including CO and Cl$^-$ (Figure S9), also showed higher concentration during mid-heating, supporting our speculation (Luo et al., 2019;Xiao et al., 2022). While the increase in $NO_2$ could be attributed to enhanced biomass burning during mid-heating, the concentration of $K^+$, primarily a biomass burning marker (Xiao et al., 2024), exhibited slight variation during the three periods and was significantly lower than Cl$^-$ (Table 2). Thus, the impact of coal combustion heating on $NO_3^-$ sources was evident, despite mid-heating periods being unfavorable for $NO_3^-$ generation.

***Question 3****: Lines 381-382: As previously mentioned, given that the Coal Replacement Project has led to more natural gas usage in the recent year (Lines 356-357), I suspect natural gas combustion might be an important $NO_x$ source in your area too. Also, in a previous study, it was reported that $NO_x$ emissions driven by natural gas (i.e., liquid-fuel) combustion are one of the important $NO_x$ sources in urban areas, accounting for a larger portion than soil and biomass burning (e.g., Bekker et al., 2023). Please clarify why four sources except for natural gas specifically are chosen for the mixing model run (Line 381-382).*

**Response 3:** Thanks for the comment. We extend our sincerest apologies for any confusion caused by our lack of clarity, which has resulted in a misunderstanding on your part. Only four sources are considered here, mainly for comparison with previous studies and thus to emphasize the importance of isotopic fingerprint of $NO_x$ sources in a region-specific context. Indeed, our findings substantiate our hypothesis that there are discrepancies in the outcomes of model resolution utilizing the $\delta^{15}$N-$NO_x$ values of disparate sources reported abroad in comparison to those established locally (Figure 3). In addition, we also assess the contribution of natural gas

sources in the following. It was found that neglecting the contribution of natural gas sources may lead to a lack of clarity in our understanding of the sources of nitrate in Tianjin.

L442-451: Previous studies have employed the MixSIAR model to estimate the contribution of $NO_x$ sources, utilizing the known $\delta^{15}N$ values of $NO_x$ in diverse sources, primarily sourced from overseas research (Zong et al., 2017;Zong et al., 2020b;Zhao et al., 2020;Zhang et al., 2020). Additionally, these studies have focused on four primary sources of $NO_x$, namely coal combustion, biomass combustion, vehicle exhaust, and soil sources. Therefore, in order to compare with previous studies, this study have focused on above four sources of $NO_x$: coal combustion, biomass combustion, vehicle exhaust, and soil sources. That is, the $\delta^{15}N$ data of above four sources from previous studies (Scenario 1) and this study (Scenario 2) were input into the MixSIAR model to quantify the sources of $NO_3^-$, respectively (Table 1).

*Question 4: Lines 386-389: The author argued that if the mixing model is run for the entire sampling period, the results for scenarios 1 and 2 are insignificantly different. However, the results for scenario 1 are slightly lower than those for scenario 2 during the pre-heating periods. This part is confusing and there is no reported statistical analysis. Considering the larger uncertainties, especially for scenario 2, it is likely that the difference is not significant, but this should be more thoroughly addressed. In addition, what does it mean if the scenarios do not produce different results? How can you state that scenario 2 leads to fewer inaccuracies (Lines 398-399) through the mixing model results?*

**Response 4:** Thanks for the comment. We apologies for any confusion caused by our lack of clarity. We have reorganized the logic of this section in the revised manuscript. Note that we did not perform statistical analyses as the model outputs were averaged. Therefore, we have corrected some inappropriate expressions in the revised manuscript. Nevertheless, the 4.9% overestimation of the contribution of Case 1 to vehicle emissions in comparison to Case 2 during the pre-heating period is evident. Considering that most of the source data of Scenario 1 originated from foreign studies (Table 1), the Tianjin source data of Scenario 2 is more representative of its localized characteristics. Therefore, we conclusion that the calculation of the contributions of various sources to $NO_3^-$ using $\delta^{15}N$ data in $NO_x$ sources from previous studies may result in inaccuracies. In addition, we have also added a discussion of the uncertainty of the results of the two scenarios in the subsequent discussion. It was found that the contributions of certain sources in Scenario 2 remained relatively stable, suggests that the results of Scenario 2 are more reliable than those of Scenario 1. However, a significant negative correlation between the PDFs of vehicle emissions and coal combustion and soil sources both in both Scenario 1 and Scenario 2 (Table S2), indicating that these sources cannot be completely differentiated (Lin et al., 2024). Therefore, we added the effect of natural gas combustion sources in the subsequent discussion. In addition, the reviewers questioned the lack of difference in mean results between the two Scenario throughout the sampling period. This is mainly influenced by the offset between the differences in the results of the two Scenario for the different sampling periods. For instance, the contributions of biomass burning in Scenario 1 are underestimated by 1.5% and overestimated by 1.6% during pre-heating periods and mid-heating periods, respectively, in comparison to Scenario 2 (Figure 3a and 3b).

L452-467: Throughout the entire sampling duration, the average contributions estimated by the MixSIAR model exhibited no substantial disparities between Scenarios 1 and 2 (Figure 3a and 3b),

suggesting that localized $\delta^{15}N$ data acquisition for $NO_x$ sources might be superfluous. However, the contributions of individual $NO_x$ sources to $NO_3^-$ in $PM_{2.5}$ were found to be large different in Scenario 1 and Scenario 2 during a certain sampling period. For instance, the contributions of vehicle exhaust during the pre-heating periods exhibited a notable discrepancy, i.e., Scenario 1 (35.1 ± 22.8%) is overestimated by 4.9% compared to the Scenario 2 (30.2 ± 21.1%). Similar difference also can be found in other sampling periods. Specifically, the contributions of vehicle exhaust in Scenario 1 are overestimated by 1.3% and underestimated by 3.8% during late heating periods and mid-heating periods, respectively, in comparison to Scenario 2 (Figure 3a and 3b). Considering that most of the source data of Scenario 1 originated from foreign studies (Table 1), the Tianjin source data of Scenario 2 is more representative of its localized characteristics. Therefore, the calculation of the contributions of various sources to $NO_3^-$ using $\delta^{15}N$ data in $NO_x$ sources from previous studies may result in inaccuracies.

L468-487: The uncertainty index ($UI_{90}$), derived from posterior distribution data, serves as a metric to evaluate the uncertainty in the results calculated by the MixSIAR model (Zhang et al., 2024). A low $UI_{90}$ value indicates a low degree of uncertainty, which suggests that the results of the source contribution were stable (Shang et al., 2020). As shown in Figure 4, the $UI_{90}$ values of coal combustion and biomass combustion were lower in Scenario 1 than in Scenario 2, indicating that the results in Scenario 1 were relatively stable. However, the contributions of vehicle exhaust and soil sources in Scenario 2 were relatively stable, as their $UI_{90}$ values were lower in Scenario 2 than in Scenario 1. It can therefore be observed that the uncertainty in contributions from different sources exhibited a variety of degrees of variability that were influenced by the differing end-member values inputted into the model. Generally, the correlation of PDFs between different sources may provide insight into the validity of model calculations (Parnell et al., 2010). For instance, if the two sources cannot be completely differentiated by the model, their correlation in PDFs will exhibit a strong negative correlation (Lin et al., 2021). The study revealed a significant negative correlation between the PDFs of vehicle emissions and coal combustion and soil sources both in both Scenario 1 and Scenario 2 (Table S2), indicating that these sources cannot be completely differentiated. Therefore, the inclusion of additional sources is recommended to enhance the accuracy of estimates provided by the MixSIAR model (Lin et al., 2021).

*Question 5: Lines 420-430: Again, it was mentioned that natural gas combustion could not be negligible in this area and the measured $\delta^{15}N(NO_x)$ values from natural gas combustion were compared with the previous study. It would be helpful if you added more explanation about why your area shows more negative values compared to the previous study. Also, please add the referred values or table 1 reference in the main text.*

**Response 5:** Thanks for the comment. First, we added to the revised manuscript why natural gas combustion sources were considered. In addition, we further elaborate on why the $\delta^{15}N\text{-}NO_x$ values of natural gas combustion emissions measured in this study was compared to the results of previous studies. Also, we explained the reason why the $\delta^{15}N\text{-}NO_x$ values of natural gas combustion emissions in Tianjin were negative compared to the results of previous studies. Finally, we also added reference values of the $\delta^{15}N\text{-}NO_x$ values for natural gas combustion in the revised manuscript.

L488-509: Since the initiation of the Coal Replacement Project in 2017, the contribution of natural

gas combustion to $NO_3^-$ may not be negligible in recent years in Tianjin (Meng et al., 2022;Wang et al., 2022). For instance, Multi-resolution emission inventory for China shown that annual $NO_x$ emissions from natural gas combustion increased from $0.6 \times 10^5$ t in 2013 to $0.7 \times 10^5$ t in 2020 (Lin et al., 2024). However, previous studies have seldom examined the role of natural gas combustion in contributing to $NO_3^-$ in $PM_{2.5}$, due to limited availability of reported of $\delta^{15}N$ values of $NO_x$ resulting from natural gas combustion (Zong et al., 2022b;Walters et al., 2015b). Previous study has found that when only four sources are considered in the MixSIAR model, there is more misclassification between the contributions of any two sources (Lin et al., 2024). Furthermore, these studies demonstrate that natural gas combustion may represent a potential source of $NO_3^-$ in $PM_{2.5}$. However, as mentioned in Section 3.1 in this study, the $\delta^{15}N$-$NO_x$ values (Table 1) emitted from the combustion of natural gas in Tianjin (-24.8 ± 5.6‰) are more negative than previous reported in foreign countries (-16.5 ± 1.7‰) due to the effects of combustion efficiency and differences in combustion temperatures, among others. Therefore, in order to clarify the necessary for identify isotopic fingerprint in a region-specific context, the following comparative analyses have been conducted. That is, we refer to the $\delta^{15}N$-$NO_x$ end-member values from natural gas combustion obtained from previous studies (Scenario 3) and locally acquired in Tianjin (Scenario 4) to calculate the relative contribution fractions of the five $NO_x$ sources using the MixSIAR model (Figure 3c and 3d).

*Question 6: The Figure 3 is missing in the manuscript. Also, please clarify the x-axis of the graphs.*

**Response 6:** Thanks for the comment. The mistake has been rectified, and the x-axis of the graphs has been clarified.

L52-455: Throughout the entire sampling duration, the average contributions estimated by the MixSIAR model exhibited no substantial disparities between Scenarios 1 and 2 (Figure 3a and 3b), suggesting that localized $\delta^{15}N$ data acquisition for $NO_x$ sources might be superfluous.

L503-509: Therefore, in order to clarify the necessary for identify isotopic fingerprint in a region-specific context, the following comparative analyses have been conducted. That is, we refer to the $\delta^{15}N$-$NO_x$ end-member values from natural gas combustion obtained from previous studies (Scenario 3) and locally acquired in Tianjin (Scenario 4) to calculate the relative contribution fractions of the five $NO_x$ sources using the MixSIAR model (Figure 3c and 3d).

L510-511: In contrast to the findings of the four sources (Scenario 1 and Scenario 2), large discrepancies exist between Scenario 3 (Figure 3c) and Scenario 4 (Figure 3d).

L529-530: In both scenarios, the contribution of natural gas combustion to $NO_3^-$ was close to or even exceeds that of soil sources (Figure 3).

[Figure]

Figure 3 Comparison of fractional contributions of $NO_3^-$ sources in $PM_{2.5}$ in Tianjin estimated by different $\delta^{15}N$ values of $NO_x$ sources. The results of Scenario 1 and Scenario 3 were estimated using the $\delta^{15}N$ values of four and five $NO_x$ sources obtained from previous studies, while the results of Scenario 2 and Scenario 4 were estimated using the $\delta^{15}N$ values of four and five $NO_x$ sources obtained from this study. Also, SE = soil emission, CC = coal combustion, BB = biomass burning, VE = vehicle emission, and CG = combustion of natural gas.

***Question 7****: Lines 444-454: The mixing model was run for different scenarios, with scenarios 3 and 4 using different $\delta^{15}N(NO_x)$ values of natural gas combustion (i.e., scenario 3 is the previous values from other study and 4 is the measured value in this study). After that, the author compared scenario 3 and 4 results from the mixing model (Line 431-448) and made a conclusion that natural gas combustion is important for $NO_3^-$ formation in this area. This part lacks an explanation in drawing the conclusion. Please clarify why natural gas is important from the two scenario results even though they account for a smaller portion than the other sources.*

**Response 7:** Thanks for the comment. We have further justified why it is important to consider natural gas combustion in the revised manuscript. On the one hand, the exclusion of natural gas combustion may directly result in the contribution fraction of other sources being overestimated by the model by more than 16%. On the other hand, the contribution of various $NO_x$ sources was becoming more stable, and the inter-influence between various sources reduced when the natural gas combustion input was introduced into the MixSIAR model.

L510-545: In contrast to the findings of the four sources (Scenario 1 and Scenario 2), large discrepancies exist between Scenario 3 (Figure 3c) and Scenario 4 (Figure 3d). Especially the contribution fractions of natural gas combustion (Scenario 3: 21.0 ± 13.8%, Scenario 4: 16.5 ±

11.5%) and coal combustion (Scenario 3: 18.2 ± 10.7%, Scenario 4: 22.0 ± 12.7%), the results estimated in Scenario 4 significantly differ from those in Scenario 3. These disparities are also present across different sampling periods. During pre-heating periods, contributions of vehicle exhaust (Scenario 3:24.9 ± 18.5%, Scenario 4: 25.6 ± 19.0%) and biomass burning (Scenario 3: 20.9 ± 15.1%, Scenario 4: 24.1 ± 17.2%) were lower in Scenario 4 compared to Scenario 3. Conversely, natural gas combustion (Scenario 3: 21.5 ± 14.3%, Scenario 4: 17.3 ± 11.3%) and soil sources (Scenario 3: 14.4 ± 9.7%, Scenario 4: 13.2 ± 8.7%) estimates in Scenario 3 were higher than those in Scenario 4. Similar differences were observed during the mid-heating periods. However, in the late-heating periods, contributions of vehicle exhaust (Scenario 3: 22.1 ± 18.0%, Scenario 4: 26.0 ± 17.3%) and coal combustion (Scenario 3: 15.6 ± 10.8%, Scenario 4: 18.2 ± 11.6%) calculated in Scenario 4 was higher than those in Scenario 3. In addition, biomass burning (Scenario 3: 20.8 ± 14.9%, Scenario 4: 20.6 ± 14.9%), natural gas combustion (Scenario 3: 24.1 ± 16.1%, Scenario 4: 18.7 ± 13%) and soil sources (Scenario 3: 17.4 ± 10.8%, Scenario 4: 16.4 ± 9.9%) in Scenario 4 were lower than those in Scenario 3. In both scenarios, the contribution of natural gas combustion to $NO_3^-$ was close to or even exceeds that of soil sources (Figure 3). This implies that the exclusion of natural gas combustion may directly result in the contribution fraction of other sources being overestimated by the model by more than 16%. Moreover, the correlation of PDFs between any two sources decreased in Scenario 4 than in other Scenario (Table S2). It can be inferred that the inter-influence between these sources diminished in Scenario 4 (Lin et al., 2024). On the other hand, the observed decline in $UI_{90}$ values of all sources when the natural gas combustion input was introduced into the MixSIAR model indicates that the result is a relatively stable calculated outcome (Figure 4) (Zhang et al., 2024). Overall, the performance of the SIAR model for Scenario 4 was much better than other Scenarios. This underscores the need to consider natural gas combustion when assessing $NO_3^-$ sources in $PM_{2.5}$, particularly in urban areas impacted by the Coal Replacement Project (Zhang et al., 2024;Lin et al., 2024). Consequently, our result further highlights that the natural gas combustion as a source input the model could improve the validity of the calculations to a certain extent. Additionally, measuring the $\delta^{15}N$ values of the local $NO_x$ source is necessary to accurately identify the source of $NO_3^-$ in $PM_{2.5}$.

*Question 8: Lines 462-465: how is the $\delta^{15}N(NO_x)$ value from the iron and steel industry distinct from other sources? And why? In Figure 1, $\delta^{15}N(NO_x)$ from the industrial emission shows the largest range, encompassing all the values. Please clarify how you can differentiate this value from others.*

**Response 8:** Thanks for the comment. Based on the LSD test, there was a significant difference between the various sources of $\delta^{15}N$-$NO_x$ signature (Figure S5). Generally, the combustion sources can produce both thermal and fuel $NO_x$. Thermal $NO_x$ is generated at high temperatures exceeding 1500°C and is influenced by factors such as the molar concentrations of $O_2$ and $N_2$ and combustion temperature (Walters et al., 2015b). In contrast, fuel $NO_x$ is primarily related to the nitrogen content of the fuel (Walters et al., 2015b). It should be noted that the raw materials of steel industry used for sintering were iron ore fines and coke powder, which differed from those used in coal combustion in power plants. Consequently, the $\delta^{15}N$ values of $NO_x$ released cannot be considered to be the same source isotopic fingerprint, as they are influenced by differences in [15]N

abundance and combustion temperature (Heaton, 1990). There is an area of overlap between the $\delta^{15}$N-NO$_x$ data from industrial emissions and that from natural gas combustion (Figure). This is primarily attributable to the intricate and multifaceted mechanisms through which NO$_x$ is produced by steel smelting apparatus or procedures. Overall, however, the mean $\delta^{15}$N-NO$_x$ value of industrial emissions was much higher than in natural gas combustion. This suggest that they are two independently existing source signal features. Of course, more work needs to be done in the future to explore in depth and refine the $\delta^{15}$N-NO$_x$ values from these sources.

[Figure]

Figure S5 Comparison of the significance of differences between various sources of $\delta^{15}$N-NO$_x$ values. The p-values at the top of the rectangular boxes indicate significant differences between different data based on Fisher's least significant difference (LSD) test.

L283-306: NO$_x$ emissions from industrial sources, such as the iron and steel industry, arise from various processes including sintering, pelletizing, and hot blast furnaces (Wang et al., 2019;Zhao et al., 2017). Generally, the $\delta^{15}$N-NO$_x$ value from industrial emission sources differs significantly from those of emissions from fossil fuel combustion (Figure S5), as indicated by the LSD test. This emphasizes the representativeness of the isotopic fingerprint in industrial emission sources. It should be noted, however, that there is an area of overlap between the $\delta^{15}$N-NO$_x$ data from industrial emissions and that from natural gas combustion (Figure S5). This is primarily attributable to the intricate and multifaceted mechanisms through which NO$_x$ is produced by steel smelting apparatus or procedures. For instance, the $\delta^{15}$N values of NO$_x$ emitted from the hot blast furnace were −43.1 ± 12.3 ‰, in contrast to the more positive value observed for the sintering

process ($-14.5 \pm 3.2‰$) and the pelletizing process ($-6.4 \pm 2.5‰$) (Figure S6). The maximum temperature in the central area of the hot air stove can exceed 2000 °C, with the majority of emitted $NO_x$ being thermal $NO_x$ (Toof, 1986). Due to the continuous $^{14}N^{14}N$ supplementation, generated $NO_x$ exhibits a negative $\delta^{15}N$. In contrast, the temperatures of sintering and pelletizing processes are relatively low (1200 ~ 1400 °C), with the majority of emitted $NO_x$ being fuel-type $NO_x$ (Toof, 1986). Specifically, functional groups such as pyrrole and pyridine in coke powder decompose at high temperatures and react with $O_2$ to produce $NO_x$, resulting in a positive value of $\delta^{15}N$-$NO_x$ (Hayhurst and Vince, 1980). It should be noted that the raw materials used for sintering were iron ore fines and coke powder, which differed from those used in coal combustion in power plants. Consequently, the $\delta^{15}N$ values of $NO_x$ released cannot be considered to be the same source isotopic fingerprint, as they are influenced by differences in $^{15}N$ abundance (Heaton, 1990).

L553-559: Our investigation has revealed that the $\delta^{15}N$-$NO_x$ signature from the iron and steel industry is distinct from that of other sources, such as vehicle exhaust, coal combustion, and biomass burning (Figure S5). The discrepancy was primarily attributed to variations in the $^{15}N$ abundance of the fuel and the combustion technology employed, among other factors, as discussion can be found in section 3.1 in this study. Consequently, it is necessary to treat this source as a unique end-member in the apportionment of $NO_3^-$.

***Question 9***: *Line 198: it was mentioned that the nitrogen isotope fractionation coefficient during $NO_x$ to $NO_3^-$ conversion is calculated in Text S2, but it is unclear how these calculations and final values are applied to the $\delta^{15}N$ values for the mixing model input.*

**Response 9:** Thanks for the comment. we have added a discussion of isotope fractionation to the revised manuscript.

L410-438: As shown in Figure 2c, the $\delta^{18}O$-$NO_3^-$ values in this study ranged from 48.3‰ to 102.9‰, with a mean $\delta^{18}O$ value of $81.1 \pm 11.5‰$ (Table 2). Similar to $\delta^{15}N$-$NO_3^-$ values, the most positive $\delta^{18}O$-$NO_3^-$ value was observed during mid-heating ($89.8 \pm 9.9‰$), followed by pre-heating ($84.5 \pm 8.4‰$) and late-heating ($73.0 \pm 9.8‰$) (Table 2). Furthermore, a significant positive linear correlation ($r = 0.7$, $p < 0.01$) was identified between $\delta^{15}N$-$NO_3^-$ and $\delta^{18}O$-$NO_3^-$, indicating that only two predominant oxidation pathways (•OH and $N_2O_5$ hydrolysis) govern $NO_3^-$ formation in this study (Xiao et al., 2020;Walters and Michalski, 2016). Previous studies have shown that the $\delta^{15}N$-$NO_3^-$ values in $PM_{2.5}$ does not fully reflect the initial $\delta^{15}N$-$NO_x$ due to the fractionation process between $NO_x$ and $NO_3^-$ (Fan et al., 2020;Song et al., 2019). Therefore, we calculated the initial $\delta^{15}N$-$NO_x$ values based on the $\delta^{15}N$ and $\delta^{18}O$ values of $NO_3^-$ as follows (Zong et al., 2017). First, the relative contributions of the •OH and $N_2O_5$ pathway were calculated separately using the $\delta^{18}O$-$NO_3^-$ values of $PM_{2.5}$. Second, the corresponding εN values ($\varepsilon_{•OH}$ and $\varepsilon_{N2O5}$) for the •OH and $N_2O_5$ pathways were estimated by considering the equilibrium isotopic fractionation between $NO_2$ and NO, and between $N_2O_5$ and $NO_2$, respectively. Finally, the $\varepsilon N$ value of $NO_x$ to $NO_3^-$ in $PM_{2.5}$ was calculated using the contributions of the two pathways and their corresponding $\varepsilon_{•OH}$ and $\varepsilon_{N2O5}$ values. The detailed procedures for all calculations can be found in the Supporting Information (Text 2). As shown in Figure S11, the contributions of •OH and $N_2O_5$ pathways were $35.4 \pm 19.8\%$ and $64.6 \pm 19.8\%$, respectively, suggesting that $N_2O_5$ pathways dominates $NO_3^-$ formation. However, the contributions varied across different sampling periods, indicating that the pathway for $NO_3^-$ formation also varied. This finding aligns with the

results presented in section 3.2.1. The calculated $\varepsilon N$ value of $NO_x$ to $NO_3^-$ were $7.5 \pm 3.4‰$ (Figure S12), close to the results of the previous studies in Beijing (Fan et al., 2020;Song et al., 2019), a large municipality near Tianjin. Furthermore, a slight difference in the $\varepsilon N$ value was found during different sampling periods (Figure S12), further indicating that isotopic fractionation similarly affects the feedback of $\delta^{15}N$-$NO_3^-$ to the $NO_x$ source.

**Minor comments:**

*Question 1: Line 2: Please make the subscript x for NOx in the entire manuscript and supplementary.*

**Response 1:** Thanks for the comment. The x for NOx is all revised to subscripts in the entire manuscript and supplementary.

*Question 2: Lines 42-44: Please check the sentence.*

**Response 2:** Thanks for the comment. The sentence has been rewrite.

L41-44: A series of scientific and effective measures to control air pollutant emissions has resulted in a notable reduction in the concentration of $SO_4^{2-}$ in $PM_{2.5}$ in urban areas of China (Wang et al., 2022).

*Question 3: Lines 58-62: Please make clear these lines since however are repeated in every sentence.*

**Response 3:** Thanks for the comment. These sentences have been rewrite.

L55-64: The reliable identification of the sources of $NO_x$ in the atmosphere was achieved using the stable nitrogen isotopes composition ($\delta^{15}N$) (Zong et al., 2017;Song et al., 2021). Furthermore, to accurate identification of $NO_3^-$ sources, it is essential to accurately identify the $\delta^{15}N$ values of the atmospheric $NO_x$ source (Zhang et al., 2024;Lin et al., 2021). It should be noted that the $\delta^{15}N$ values from various $NO_x$ sources have been reported in previous studies (Zong et al., 2020a;Zong et al., 2022a). However, these $\delta^{15}N$ values of $NO_x$ in different sources mainly from foreign countries, and the collection methods have not been unified (Elliott et al., 2019;Walters et al., 2015a;Walters et al., 2015b), resulting in some variation in $\delta^{15}N$ values of $NO_x$ from the same source.

*Question 4: Lines 123-124: I would suggest adding a detailed site description in this part, including information on the population of Tianjin and its size, to provide a better understanding of the area.*

**Response 4:** Thanks for the comment. We have added a detailed site description in this part in the revised manuscript.

L128-134: Tianjin is a representative megacity situated in the eastern portion of the North China Plain (Xiao et al., 2024). As of the end of 2018, the permanent population of Tianjin had reached

16 million, representing a growth rate of 27,300 compared to the previous year (National Breau of Statistics of China, 2018). The city occupies an area of approximately 11,919.7 square kilometers, with agricultural land accounting for 6,921.4 of those square kilometers and arable land representing 4,367.6 square kilometers (National Breau of Statistics of China, 2018).

*Question 5: Lines 132-135: It would be helpful to mark the monitoring stations on the map.*

**Response 5:** Thanks for the comment. We have added it. See in particular the lower right-hand panel in figure S1.

[Figure]

Figure S1 Map of PM$_{2.5}$ and NO$_x$ source sampling locations

*Question 6: Line 430: Can you check if the figure number is correct? Can't find (d) in Figure 4.*

**Response 6:** Thanks for the comment. This error has been corrected in the revised manuscript.

Therefore, we refer to the $\delta^{15}$N-NO$_x$ end-member values from natural gas combustion obtained from previous studies (Scenario 3) and locally acquired in Tianjin (Scenario 4) to calculate the relative contribution fractions of the five NO$_x$ sources using the MixSIAR model (Figure 3c and 3d).

*Question 7: Lines 431-446: It is quite hard to compare which fraction values represent scenarios 3 and 4 in parentheses. Please clarify these values.*

**Response 7:** Thanks for the comment. We apologize for the confusion caused by our oversight. We have improved this section in the revised manuscript.

L510-530: In contrast to the findings of the four sources (Scenario 1 and Scenario 2), large discrepancies exist between Scenario 3 (Figure 3c) and Scenario 4 (Figure 3d). Especially the contribution fractions of natural gas combustion (Scenario 3: 21.0 ± 13.8%, Scenario 4: 16.5 ± 11.5%) and coal combustion (Scenario 3: 18.2 ± 10.7%, Scenario 4: 22.0 ± 12.7%), the results

estimated in Scenario 4 significantly differ from those in Scenario 3. These disparities are also present across different sampling periods. During pre-heating periods, contributions of vehicle exhaust (Scenario 3:24.9 ± 18.5%, Scenario 4: 25.6 ± 19.0%) and biomass burning (Scenario 3: 20.9 ± 15.1%, Scenario 4: 24.1 ± 17.2%) were lower in Scenario 4 compared to Scenario 3. Conversely, natural gas combustion (Scenario 3: 21.5 ± 14.3%, Scenario 4: 17.3 ± 11.3%) and soil sources (Scenario 3: 14.4 ± 9.7%, Scenario 4: 13.2 ± 8.7%) estimates in Scenario 3 were higher than those in Scenario 4. Similar differences were observed during the mid-heating periods. However, in the late-heating periods, contributions of vehicle exhaust (Scenario 3: 22.1 ± 18.0%, Scenario 4: 26.0 ± 17.3%) and coal combustion (Scenario 3: 15.6 ± 10.8%, Scenario 4: 18.2 ± 11.6%) calculated in Scenario 4 was higher than those in Scenario 3. In addition, biomass burning (Scenario 3: 20.8 ± 14.9%, Scenario 4: 20.6 ± 14.9%), natural gas combustion (Scenario 3: 24.1 ± 16.1%, Scenario 4: 18.7 ± 13%) and soil sources (Scenario 3: 17.4 ± 10.8%, Scenario 4: 16.4 ± 9.9%) in Scenario 4 were lower than those in Scenario 3. In both scenarios, the contribution of natural gas combustion to $NO_3^-$ was close to or even exceeds that of soil sources (Figure 3).

*Question 8: Line 524: What does mean "these two sources"? Please clarify this.*

**Response 8:** Thanks for the comment. These two sources are coal combustion and biomass combustion, which we have clarified in the revised manuscript.

L617-619: Additionally, the contributions of coal combustion and biomass burning were higher during pre-heating compared to late-heating periods, when six sources were considered in the MixSIAR model.

*Question 9: Lines 902-905: the description of IE is missing in the caption.*

**Response 9:** Thanks for the comment. The description of IE is industrial emissions, the description has been added in the revised manuscript.

[Figure]

Figure 5 The contribution fraction of the four, five, and six sources in different periods estimated by the isotopic fingerprint in $NO_x$ sources obtained from this study. Also, IE = Industrial emissions, SE = soil emission, CC = coal combustion, BB = biomass burning, VE = vehicle

emission, and CG = combustion of natural gas.

***Question 10****: Figure 6: (a) and (b) appear to be repetitive of Fig 5. Please revise to avoid redundancy.*

**Response 10:** Thanks for the comment. Although some of the results in Figures 6a and 6b have been shown in Figure 5. However, the results in Figs. 6a and 6b are meant to be compared with the concentrations of $K^+$ and $Cl^-$ to further illustrate the reliability of our results. Therefore, we did not remove Figures 6a and 6b.

[revised manuscript text omitted]